# Disease-specific tau filaments assemble via polymorphic intermediates

Sofia Lövestam[1], David Li[1], Jane L. Wagstaff[1], Abhay Kotecha[2], Dari Kimanius[1], Stephen H. McLaughlin[1], Alexey G. Murzin[1], Stefan M. V. Freund[1], Michel Goedert[1,3 ✉] & Sjors H. W. Scheres[1,3 ✉]

Intermediate species in the assembly of amyloid filaments are believed to play a central role in neurodegenerative diseases and may constitute important targets for therapeutic intervention[1,2]. However, structural information about intermediate species has been scarce and the molecular mechanisms by which amyloids assemble remain largely unknown. Here we use time-resolved cryogenic electron microscopy to study the in vitro assembly of recombinant truncated tau (amino acid residues 297–391) into paired helical filaments of Alzheimer's disease or into filaments of chronic traumatic encephalopathy[3]. We report the formation of a shared first intermediate amyloid filament, with an ordered core comprising residues 302–316. Nuclear magnetic resonance indicates that the same residues adopt rigid, β-strand-like conformations in monomeric tau. At later time points, the first intermediate amyloid disappears and we observe many different intermediate amyloid filaments, with structures that depend on the reaction conditions. At the end of both assembly reactions, most intermediate amyloids disappear and filaments with the same ordered cores as those from human brains remain. Our results provide structural insights into the processes of primary and secondary nucleation of amyloid assembly, with implications for the design of new therapies.

The assembly of amyloid-β, tau, α-synuclein and TDP-43 (TAR DNA-binding protein 43) into amyloid filaments defines most cases of human neurodegenerative disease[1]. The hypothesis that the formation of amyloid filaments causes disease is supported by the observation that mutations in the genes that encode these proteins or increase their production give rise to inherited forms of disease[2]. Moreover, cryogenic electron microscopy (cryo-EM) structures of amyloid filaments from human brains have revealed that distinct folds of tau, α-synuclein and TDP-43 define different diseases, suggesting that specific mechanisms of amyloid formation may underlie these diseases[4–12]. Nevertheless, the molecular mechanisms by which amyloid may cause neurodegeneration remain unknown.

It has been suggested that intermediate species, on-pathway to the formation of mature filaments, are main drivers of amyloid toxicity[13]. Both non-filamentous species, so-called oligomers, and filamentous intermediates, known as protofibrils, have been proposed to play a role. Intermediate species of amyloid assembly are thus an important target for therapeutic intervention. Lecanemab, an approved drug for Alzheimer's disease with a measurable reduction of cognitive decline[14], is a humanized mouse monoclonal antibody that was raised to what were thought to be protofibrils of synthetic Aβ40 peptide with the Arctic mutation[15].

Despite the interest in intermediate species of amyloid formation, little is known about their structures. Owing to their transient nature, most experimental data on oligomers and protofibrils come from in vitro assembly reactions with recombinant proteins, including amyloid-β[16,17], tau[18] and α-synuclein[19]. Most in vitro reactions yield filaments with ordered cores that are different in structure from human brain filaments, although in some cases identical substructures have been described[9,11,20]. Only for tau have in vitro assembly conditions been reported that yield filaments that are identical to those derived from human brains. Residues 297–391 (using the numbering of the longest human brain tau isoform) constitute the proteolytically stable core of paired helical filaments (PHFs) from the brains of individuals with Alzheimer's disease[21]. The tau(297–391) construct, upon shaking in phosphate buffer with magnesium chloride, forms PHFs with ordered cores that are identical to those from human brains[3,8]. The use of sodium chloride instead of magnesium chloride[3] leads to the formation of filaments with ordered cores that are identical to those extracted from the brains of individuals with chronic traumatic encephalopathy (CTE)[6].

Here we used time-resolved cryo-EM to characterize the filamentous intermediates that form during the in vitro assembly of tau into PHFs or CTE filaments. We report the formation of a common first intermediate amyloid (FIA) in both reactions, and the presence of multiple, polymorphic filamentous intermediates, with structures that depend on the reaction conditions, at later time points. Our results provide new insights into primary and secondary nucleation of tau amyloid formation that challenge existing theories and provide new avenues for therapeutic design.

[1]MRC Laboratory of Molecular Biology, Cambridge, UK. [2]Thermo Fisher Scientific, Eindhoven, The Netherlands. [3]These authors jointly supervised this work: Michel Goedert, Sjors H. W. Scheres. ✉e-mail: mg@mrc-lmb.cam.ac.uk; scheres@mrc-lmb.cam.ac.uk

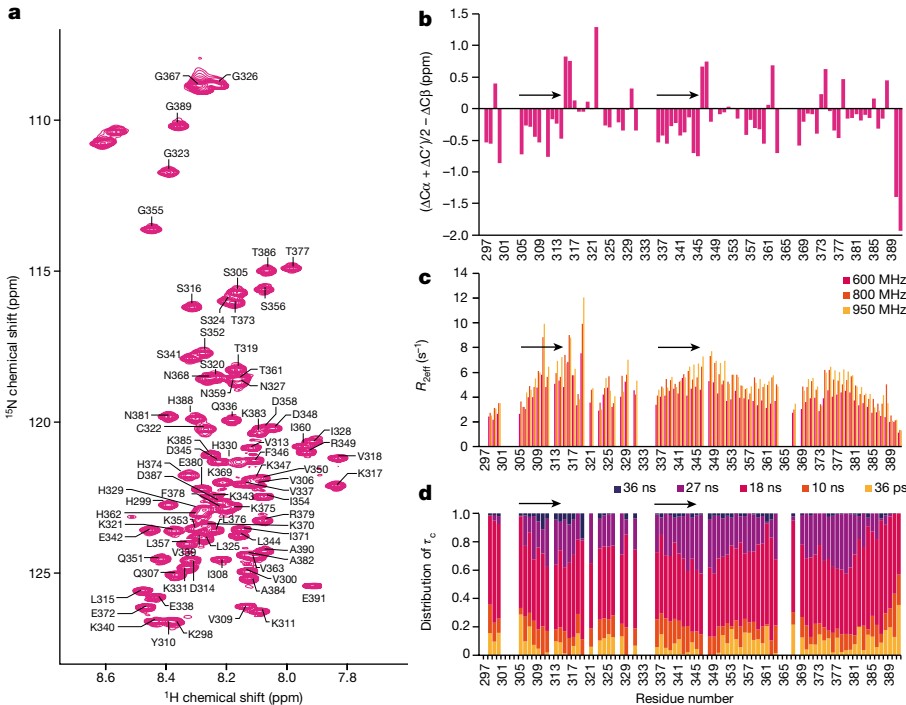

**Fig. 1 | Solution-state NMR of tau monomers. a**, Assigned 600-MHz $^{15}$N–$^1$H heteronuclear single quantum coherence spectrum of human tau(297–391). **b**, Secondary shift analysis of the backbone Cα, Cβ and C′ chemical shifts. Stretches of residues with negative values, as seen for residues 305–314 and 336–345, indicate a propensity to adopt an extended β-strand-like conformation. These residues are highlighted with black arrows in **b**–**d**. **c**, Exchange-free transverse relaxation ($R_{2eff}$) rates collected at 600 (magenta), 800 (orange) and 950 (yellow) MHz. Higher rates are indicative of increased rigidity on the millisecond–microsecond timescale. **d**, IMPACT analysis of tau motions on timescales ranging from 36 ps (yellow) to 36 ns (purple). The diagram illustrates the distribution of internal backbone motions as a distribution over five correlation times ($\tau_c$). Backbone dynamics of residues 305–317, 343–349 and 377–381 exhibit motions at slower frequencies, indicative of segmental motion associated with conformational restrictions. This is most pronounced for residues 305–317, with marked contributions from the slowest timescale of motion (36 ns).

## Parts of monomeric tau are β-strand like

We expressed and purified recombinant human tau(297–391) (Methods). Analytical ultracentrifugation indicated that at a concentration of 6 mg ml$^{-1}$, purified tau was monomeric in solution, with flexible conformations (Extended Data Fig. 1a). Solution-state nuclear magnetic resonance (NMR) confirmed the presence of disordered tau monomers and suggested that residues 305–314 and 336–345 have a tendency to adopt extended conformations reminiscent of those found in β-strands. Similar observations have also been reported for full-length 4R tau[22] and for a 4R tau construct comprising residues 244–372 (K18) or its 3R version (K19)[23,24]. Although most tau appears to be monomeric, we cannot exclude the possibility that small amounts of dimers, possibly through transient formation of intermolecular β-sheets, are present in solution too. For a more detailed analysis of the dynamic landscape of the conformational ensemble of tau(297–391), we carried out interpretation of motions by a projection onto an array of correlation times (IMPACT) analysis of backbone relaxation measurements at different field strengths[25]. IMPACT analysis indicated that motions in tau(297–391) monomers are best approximated by five correlation times ranging from 36 ps to 36 ns. In particular, three regions (residues 305–317, 343–349 and 377–381) contribute to slow segmental motion associated with the increased tendency to adopt an extended structure, which was most evident for residues 305–317 (Fig. 1 and Extended Data Fig. 1b–l).

## Tau assembly precedes thioflavin T fluorescence

We then initiated multiple replicates of two assembly reactions. The first reaction was carried out in the presence of magnesium chloride for forming PHFs, whereas the second contained sodium chloride for forming CTE filaments. To a subset of reactions, we added 1.5 μM thioflavin T (ThT) to monitor fluorescence continuously. For reactions without ThT, we took aliquots at various time points for cryo-EM structure determination. As each cryo-EM sample uses 3 μl of the 40-μl reaction, and because not all cryo-EM grids are suitable for data acquisition, we collected cryo-EM datasets from five and six replicates for each of the PHF and CTE reactions, respectively. Further replicates were used for quantification of pelletable tau by ultracentrifugation and for offline ThT monitoring, as these required the entire reaction volumes. Protein samples were prepared at multiple times to carry out the replicate experiments and the products were considered to be identical.

For both PHF and CTE reactions, continuous ThT fluorescence monitoring showed a typical sigmoidal curve that has been associated with a nucleation–polymerization model of amyloid formation[26] (Fig. 2a and Extended Data Fig. 2). For approximately the first 240 min after starting the assembly reactions, ThT fluorescence remained low, but it increased sharply between 240 and 480 min, after which it plateaued. Offline ThT measurements were in accordance with continuous monitoring, indicating that the presence of ThT in the reaction mixture did not alter the kinetics.

The samples that were used for offline ThT fluorescence measurements were also used for quantification of pelletable tau by ultracentrifugation. As abundant amyloid filaments remained in the supernatants of ultracentrifugation runs at 100,000–130,000g (refs. 27,28), we centrifuged the samples at 400,000g for 15 min at 20 °C to quantify the amount of soluble versus pelletable tau by SDS–polyacrylamide gel electrophoresis (Supplementary Fig. 53). Until 60 min, almost all tau remained soluble. However, 70–80% of tau was already pelletable at 120 min, and the amount of pelletable tau plateaued at 80–90% at 720 min (Fig. 2b).

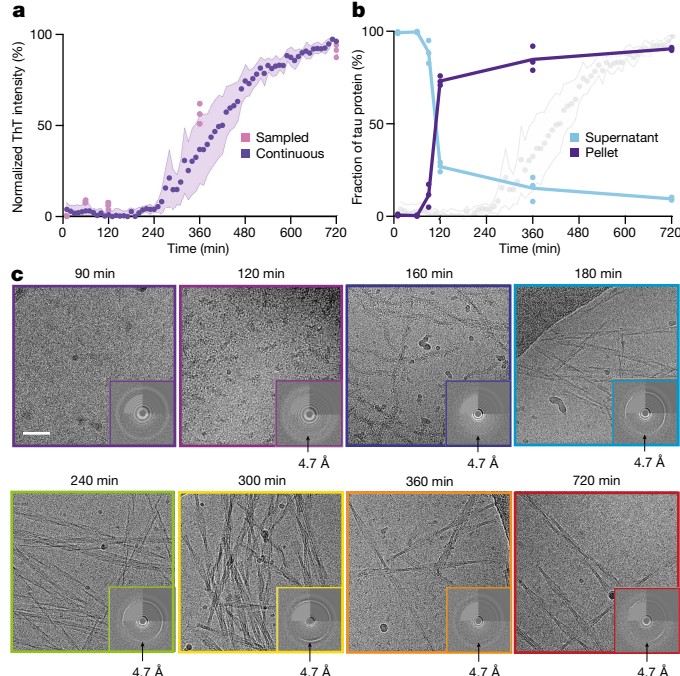

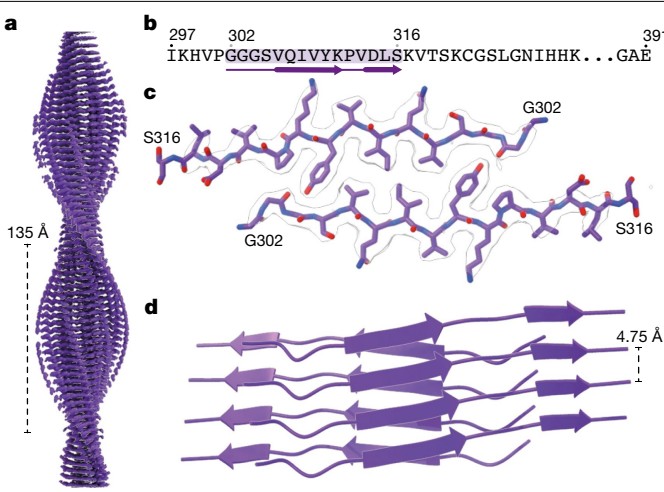

**Fig. 3 | Structure of the FIA. a**, Side view of the cryo-EM reconstruction of the FIA, with the crossover distance indicated. **b**, Amino acid sequence of the ordered core (highlighted in purple). **c**, Top view of the cryo-EM density (in transparent white) and the atomic model. **d**. Side view of the atomic model in schematic representation.

**Fig. 2 | Time-resolved cryo-EM. a**, ThT fluorescence profile of the PHF reaction. Purple circles indicate the average of three replicates of continuous ThT monitoring; purple shading indicates the standard deviation among replicates; pink circles represent individual offline ThT measurements. **b**, The amount of tau in the pellet and in the supernatant (as a percentage of the total amount of tau) after centrifugation for 15 min at 400,000*g*, quantified by SDS–polyacrylamide gel electrophoresis. **c**, Cryo-EM micrographs at various time points in the PHF reaction. Insets show the power spectrum of the electron micrographs, with the water ring at 3.6 Å and/or the 4.7 Å signal that is indicative of β-sheet structure. Scale bar, 50 nm (applies to all micrographs). The numbers of micrographs acquired for each dataset are given in the Supplementary Figs. 1–48.

Cryo-EM imaging confirmed the presence of amyloid filaments from 120 min (Fig. 2c and Extended Data Fig. 3a). Images of samples taken at 30, 60 or 90 min were devoid of filaments and did not show evidence of β-sheets. However, at 120 min, many filaments were visible in both PHF and CTE reactions, and power spectra showed a strong 4.7-Å signal, indicative of abundant β-sheets. These initial tau filaments have a fuzzy, beads-on-string-like appearance, with a short crossover distance of 13.5 nm. They range in size from just one or two crossovers to filaments longer than the field of view (about 300 nm). At later time points, numerous types of amyloid filament could be distinguished (Fig. 2c). Using helical reconstruction in RELION[29], we solved 163 cryo-EM structures from the PHF and CTE reactions. We built atomic models for 45 different structures with resolutions ranging from 1.7 to 3.8 Å (Extended Data Figs. 4 and 5, Supplementary Figs. 1–52 and Supplementary Tables 1–29).

### A transient filament assembles first

Cryo-EM structure determination revealed the presence of the same filament at 120 min in the PHF and CTE reactions (Fig. 3). As we observed no evidence of earlier filaments, and because this filament adopted a cross-β packing characteristic of amyloids, we termed it the FIA. Although filamentous, the FIA does not generate fluorescence with ThT. The FIA adopts a pseudo-$2_1$ helical symmetry and has an atypically large, left-handed twist of −6.3° (other known tau filaments, including those described in this paper have twists between −1.65° and −0.77°).

The ordered core of the FIA comprises only residues $_{302}$GGGSVQIVY-KPVDLS$_{316}$ from two antiparallel tau molecules, with a predominantly

hydrophobic close-packed interface. At its centre, the side chains of valine 306 and isoleucine 308 from opposite protofilaments pack against each other and are flanked by the side chain of tyrosine 310. Thereby, valine 306 and isoleucine 308 in the FIA form a similar tightly packed hydrophobic interface as observed in one of several crystal forms (Protein Data Bank accession code 2ON9) of the $_{306}$VQIVYK$_{311}$ peptide alone[30,31]. Whereas the β-sheets in the crystal are flat and stabilized by additional crystal contacts, β-sheets in the FIA are twisted and stabilized by additional hydrogen bonds between the hydroxyl group of tyrosine 310 to the backbone groups of glycine 303 and serine 305 (Extended Data Fig. 6).

The FIA exists only for a short time. At 120 min, 100% of the filaments that yield interpretable two-dimensional class averages are FIAs, but they are no longer observed at 160 min (Extended Data Fig. 3a). At 140 and 160 min in the PHF reaction, multiple different types of filament give rise to uninterpretable two-dimensional class averages, many of which lack helical twist (Extended Data Fig. 3b). We were unable to solve the structures of these filaments. At 160 min in the CTE reaction, we were able to solve nine structures (Extended Data Fig. 3c).

### Polymorphism in the PHF reactions

From 180 min, we observed multiple types of filament in the PHF reactions (Fig. 4a and Extended Data Fig. 7a). Most filaments at 180 min were made of two protofilaments with an ordered core that comprised residues 305–380, similar to the extent of the ordered core of PHFs[8]. As is the case of the Alzheimer fold, these protofilaments formed a turn of a β-helix at residues 337–356. However, whereas the Alzheimer fold is C shaped, the ordered cores of most protofilaments at 180 min adopted a more elongated, J-shaped conformation. In the different filament types, the J-shaped protofilaments packed against each other in different ways. During the next 3 h, additional types of filament formed. In total, we solved 24 different structures from samples taken at 120, 180, 240, 300, 360 and 720 min, with 20 maps to resolutions sufficient for atomic modelling (Extended Data Fig. 4). Again, most filaments comprised two protofilaments that packed against each other in various ways. Some filaments with three or four protofilaments also formed, including the previously described triple and quadruple helical filaments[3].

As time progressed, filaments with two J-shaped protofilaments disappeared and filaments with two C-shaped protofilaments

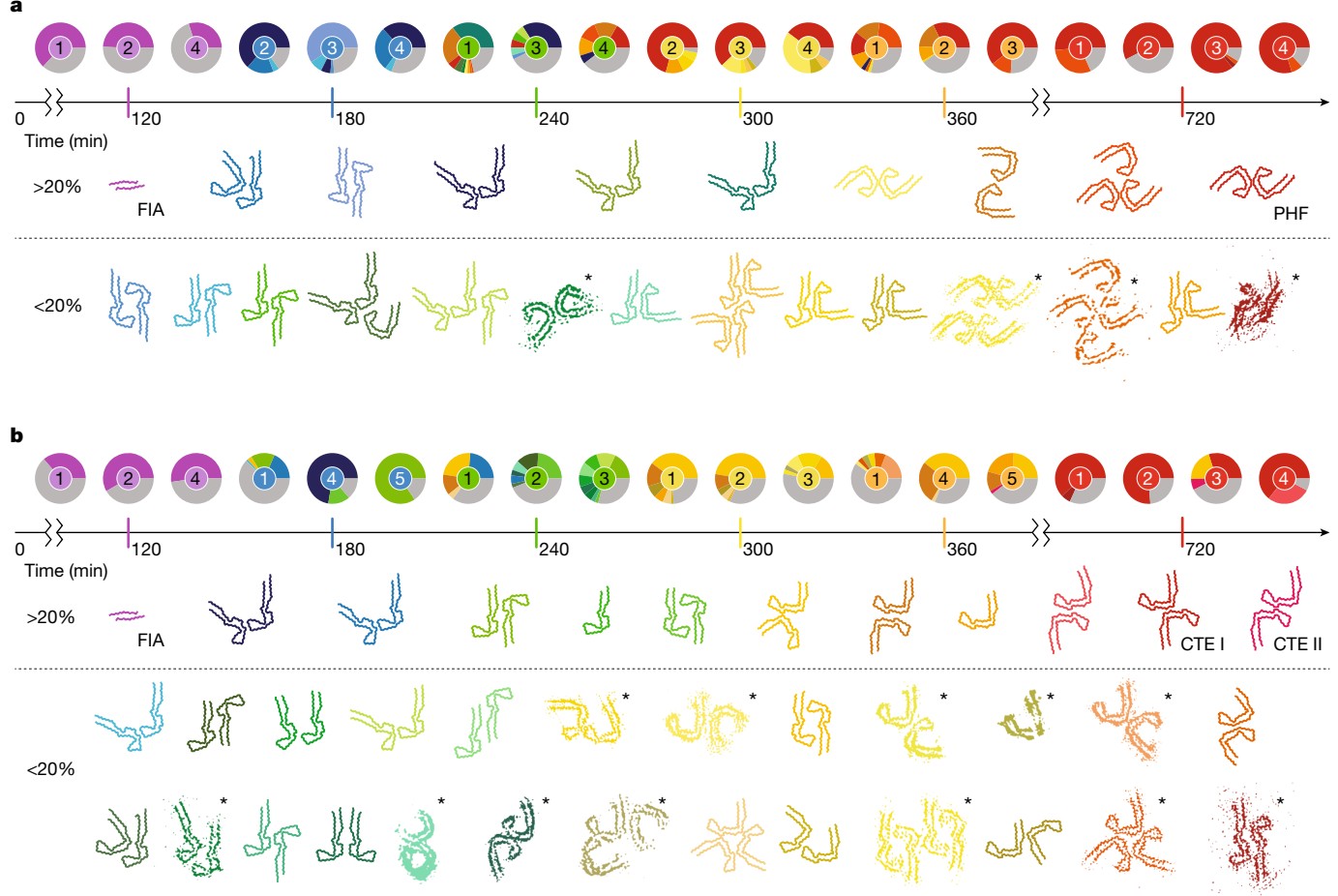

**Fig. 4 | Overview of structures in the assembly reactions. a,b,** Pie charts show the relative abundance of the structures determined for each replicate (numbered 1–5 inside the pie charts) of the PHF reaction (**a**) and the CTE reaction (**b**). Relative abundances were calculated on the basis of the distribution of particle counts from cryo-EM micrographs of each replicate. Main-chain traces for atomic structures are shown in the same colours as the pie chart segments for each reaction. Grey segments represent filaments for which no structures were solved. Structures that were solved at resolutions insufficient for atomic modelling are shown as thresholded densities and are indicated with asterisks. Structures and pie chart central circles are coloured per time point (120 min in purple; 180 min in blue; 240 min in green; 300 min in yellow; 360 min in orange and 720 min in red). All structures shown are unique and coloured according to the time point at which they are most abundant, averaged across all replicates. More abundant structures (assessed by maximal percentage across all replicates and time points) are closer to the time axis, whereas less abundant ones are further away. Details of all datasets and structures, including pie charts of additional replicates and time points, are shown in Supplementary Figs. 1–48.

appeared (Fig. 5a). Between 240 and 360 min, filaments with one J-shaped and one C-shaped protofilament were also present. The inter-protofilament packing of these filaments with one J-shaped and one C-shaped protofilament resembled the asymmetrical arrangement of protofilaments in the straight filaments extracted from the brains of individuals with Alzheimer's disease. Among J-shaped and C-shaped protofilaments, the opposing β-strands comprising residues 305–320 and 365–380 hardly changed their conformation, confining all differences to the β-helix turn and its surrounding residues. We observed two main types of J-shaped protofilament, as well as several other minority types of J-shaped protofilament (Fig. 4a). Earlier J-shaped protofilaments tend to be straighter, whereas the β-helix turn in later J-shaped protofilaments turns inwards, towards the rest of the protofilament. This change in orientation of the β-helix turn is reflected in a distinct conformation of the $_{332}$PGGG$_{335}$ motif. The difference between the later J-shaped protofilament and the earliest C-shaped protofilaments coincided with a rearrangement of the $_{364}$PGGG$_{367}$ motif on the opposite side of the protofilament. The formation of a tighter packing of residues near the $_{332}$PGGG$_{335}$ motif in the

C-shaped protofilaments compared to the J-shaped protofilaments may drive this conformational change (Extended Data Fig. 8a). Finally, the change from earlier C-shaped protofilaments to the final, more closed, C-shaped protofilaments of PHFs involves a second inwards rotation of the β-helix turn, which again concurs with a rearrangement of the $_{332}$PGGG$_{335}$ motif.

Some filaments that resembled PHFs were already present at 240 min. Although they had the same double C-shaped protofilament arrangement as in PHFs, their crossover distances tended to be more variable at earlier time points. At later time points, most filaments had crossover distances of 750–900 Å, similar to those of PHFs extracted from the brains of individuals with Alzheimer disease[8]. The amino and carboxy ends of each protofilament packed against each other within the same β-rung. However, at earlier time points, filaments with crossover distances as large as 2,900 Å formed, in which residues at the amino terminus of the protofilament packed against residues at the carboxy terminus that were one or more β-rungs lower. In addition, the position along the helical axis of the β-helix turn compared to the amino and carboxy termini of the protofilament also changed as the crossover

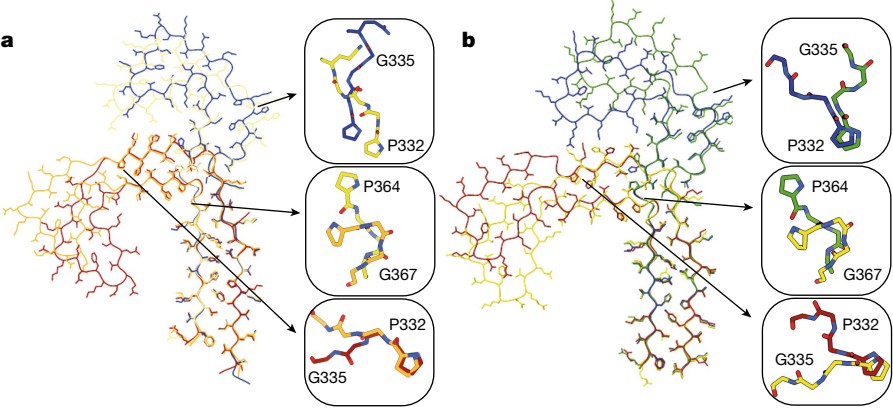

**Fig. 5 | Protofilament maturation. a**, Atomic models for protofilaments in the PHF reaction at 180 min (blue), 300 min (yellow), 360 min (orange) and 720 min (red). Insets show the corresponding conformations of the $_{322}$PGGG$_{335}$ and $_{364}$PGGG$_{367}$ motifs. **b**, As in **a**, but for the CTE reaction including a model at 240 min (green).

distances decreased. These conformational changes correlated with peptide flips at glutamic acid 342 and isoleucine 354 (Extended Data Fig. 9).

Finally, by 720 min, most filaments had adopted the same ordered core as that of PHFs extracted from the brains of individuals with Alzheimer's disease, although triple helical filaments remained in some replicates. Overall, the five replicates were relatively consistent in their timing.

### Polymorphism in the CTE reactions

In the CTE reactions, a greater number of intermediate structures formed than in the PHF reactions (Fig. 4b and Extended Data Fig. 7b). In total, we determined the structures of 40 different filament types, with 25 maps being of sufficient resolution for atomic modelling (Extended Data Fig. 5). As in the PHF reactions, most intermediate filament types consisted of two protofilaments with ordered cores that comprised residues 305–380, and the protofilaments packed against each other in multiple ways. Most filament types also adopted a J-shaped conformation at earlier time points and, as time progressed, more C-shaped protofilaments appeared. No filaments with one J-shaped and one C-shaped protofilament were observed. As for the intermediates in the PHF reaction, the $_{332}$PGGG$_{335}$ and $_{364}$PGGG$_{367}$ motifs and possibly a tighter packing in the C-shaped protofilaments appeared to play a central role in the maturation of J-shaped to C-shaped protofilaments (Fig. 5b and Extended Data Fig. 8b).

The presence of sodium chloride in the CTE reaction affected the conformation of the β-helix turn in all intermediate amyloids that formed after the FIA and the final CTE structures, which showed a more open β-helix turn than in the Alzheimer fold, together with the presence of an extra density inside the β-helix turn. This extra density was previously interpreted as sodium chloride ion pairs[2]. In the PHF reaction, some earlier intermediate filaments also showed a similar extra density. It is likely that traces of sodium chloride, which was used during purification of recombinant tau(297–391), were still present in the PHF reaction.

Most intermediates that formed in the CTE reactions comprised two identical protofilaments; some filaments made of either one protofilament or three protofilaments were also present. Compared to the filaments in the PHF reactions, intermediates in the CTE reactions exhibited a greater variation in inter-protofilament packing. Many packings seemed to be coordinated by electrostatic interactions (Extended Data Fig. 10). Relatively small differences in the protofilament packing of individual pairs of filament types suggest that intermediate amyloid filaments may mature through subsequent sliding of their protofilaments relative to each other.

After 720 min, all reactions contained CTE type I filaments[6]. In replicates 2, 3, 4 and 5, CTE type II filaments were also present. The different replicates of the CTE reaction were reasonably well synchronized, except for replicate 3, which at 720 min still contained intermediates that were present at 360 min in the other replicates.

## Discussion

Polymorphism is a common phenomenon in crystallography. Ostwald's interpretation of crystal polymorphism explains how the state that nucleates is not necessarily the most thermodynamically stable. Instead, the state that most closely resembles the solution state is kinetically advantaged[32]. This interpretation may also be relevant for understanding the assembly of tau into amyloid filaments. Being the product of a long disease process, tau PHFs and CTE filaments probably represent a thermodynamically stable state. In vitro assembly of recombinant tau(297–391) converges onto the same structures over 12 h, but only after multiple polymorphic intermediate amyloids have formed and disappeared again.

In the first intermediate, the FIA, only 15 residues of each tau molecule are ordered; the remaining 80 residues are not resolved in the cryo-EM map, suggesting that they remain largely unstructured. Thereby, for 84% of the residues in the FIA, the first detectable nucleated state probably closely resembles the solution state. Our solution-state NMR data suggest that some of the 15 residues of the FIA's ordered core may already adopt extended, β-strand-like conformations in monomeric tau, with slower dynamics than the rest of the protein, which will reduce further the differences between the solution and nucleated states. The fact that β-sheets in the FIA are more twisted than in other amyloids may also play a role. The ordered core of the FIA explains the previously observed importance of the $_{306}$VQIVYK$_{311}$ (PHF6) motif for the assembly of full-length human tau into filaments in vitro[33] and in transgenic mice[34]. The PHF6 motif is also essential for the seeded assembly of tau in transfected cells[35]. Its absence may explain why microtubule-associated protein 2 (MAP2) does not form disease inclusions[36]. The existence of filamentous intermediates in amyloid assembly reactions has been reported previously (for example, using atomic force microscopy of amylin)[37]. It remains to be seen whether primary nucleation (that is, formation of filaments that is independent of the presence of other filaments) of other proteins for which small amyloid-prone regions have been identified, such as acylphosphatase, amyloid-β, α-synuclein, TDP-43, transthyretin and the prion protein[38–41], occurs through similar FIA structures.

The sudden assembly of 70–80% of tau molecules into the FIA at 120 min, after a 90-min lag period, together with the complete

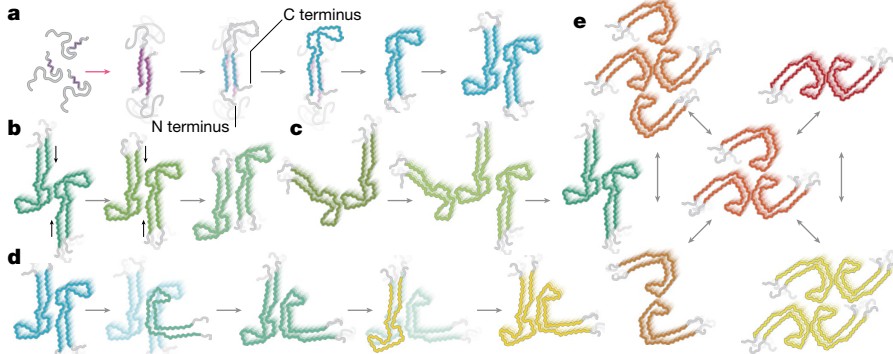

**Fig. 6 | Models for primary and secondary nucleation. a**, Disordered parts of the molecules are shown in grey; other colours represent ordered structures, with the same colours as in Fig. 4. Primary nucleation (pink arrow) of disordered monomers with partially rigid β-strands may lead to formation of the FIA. Subsequent secondary nucleation (grey arrows) may then occur through folding back of the carboxy-terminal domain of tau monomers at the end of, or at defects in, the FIA to form the interface between residues 305–316 and 370–380 that remains nearly constant in all intermediate and final filament types. This folding back may then lead to the seed of an early J-shaped protofilament. Possibly, the formation of singlets of J-shaped protofilaments (as observed in the CTE reaction) is followed by packing of a second protofilament to form more stable doublets. **b**, Secondary nucleation by sliding of protofilaments relative to each other (indicated with small black arrows), again at the ends of, or at defects in, filaments may lead to the subsequent formation of more stable structures. **c**, Alternatively, secondary nucleation through the addition and removal of protofilaments may happen when more stable protofilament interactions form at the sides of existing filaments. **d**, Secondary nucleation through conformational change may lead to protofilament maturation, from early J-shaped protofilaments to later C-shaped protofilaments, again at the end of, or at defects in, filaments. **e**, Secondary nucleation through the splitting of filaments may lead to two smaller filament types (for example, Extended Data Fig. 3f). Combined with secondary nucleation through the addition and removal of protofilaments, this may lead to an interplay between multiple filament types.

disappearance of the FIA after 180 min, suggests that primary nucleation of the FIA is rate-limiting and that, once formed, the FIA can seed the rapid growth of other filament types through mechanisms of secondary nucleation (that is, formation of filaments that depends on the presence of other filaments). On the basis of kinetic studies[42,43], secondary nucleation has been proposed to play a central role in amyloid assembly, but molecular insights into how such nucleation processes happen have been scarce. Our cryo-EM structures provide a basis for new hypotheses on how such secondary nucleation processes may occur (Fig. 6).

Whereas residues 302–316 from both protofilaments of the FIA pack in a homotypic, antiparallel manner, residues 305–316 pack in a heterotypic arrangement against residues 370–380 within the same molecule in later filament types. Therefore, major topological rearrangements need to occur in the conversion of the FIAs into any of the later intermediates. The observation that residues 305–316 and 370–380 adopt almost the same conformation in all later intermediates and the final structures suggests that this part of the protofilament forms early and is relatively stable. Possibly, residues 370–380 from one tau molecule in the FIA fold back and pack against residues 305–316 of the same molecule, instead of forming the homotypic packing with another tau molecule in the opposite protofilament. This could happen at either end of the FIAs and at defects along their length. Once the seed for a J-shaped protofilament forms in this way, it may then lead to the formation of various filament types with two J-shaped protofilaments (Fig. 6a). Other models of secondary nucleation include: maturation of filament types through the sliding of protofilaments relative to each other (Fig. 6b); growth, association and/or dissociation of protofilaments at the sides of filaments (Fig. 6c,e); and formation of new filament types by partial structural templating at the end of, or at defects in, existing protofilaments (Fig. 6d). It is possible that any of these mechanisms also lead to the conversion of one filament type into another. Filaments with variations in structure along their length have been observed previously (for example, for heparin-induced filaments of recombinant tau[44], for TDP43 filaments from human brain[4] and for filaments of immunoglobulin light chains from human heart[45]). In our images, most filaments are not branched and do not appear to consist of multiple filament types, although we do observe some exceptions (Extended Data Fig. 3d–g).

These cryo-EM snapshots do not provide insights into the kinetics of the assembly reactions. The FIA, the later intermediates and the PHF and CTE filaments are probably in exchange with monomers from the solution, leading to their continuous elongation and disassembly. Future kinetic studies are required to explore in detail the interplay between species. Our results indicate that ThT fluorescence is suboptimal for the quantification of filament formation, as many intermediate amyloids do not give rise to ThT fluorescence. Amyloid filaments that do not give rise to ThT fluorescence have been reported previously (for example, for TDP43 (ref. 46), as well as for transthyretin, $\beta_2$-microglobulin and some forms of amyloid-$\beta$[47]). The factors that determine whether an amyloid filament is detected by ThT fluorescence remain unclear.

In human brains, tau pathology is thought to spread through templated seeding, akin to the spread of prions, for which small amounts of seeds accelerate the assembly of monomeric protein[48]. In disease, the initial seeds may form through the same short-lived intermediates as described here, but it will be difficult to prove their presence in human brains. The molecular mechanisms of templated seeding remain poorly understood. On the one hand, growth of filaments at their ends alone does not explain the kinetics of seeded aggregation in vitro[49]. On the other hand, it is not clear how structural templating happens in alternative models, in which secondary nucleation has been proposed to happen on the sides of filaments[43]. Seeded assembly in vitro does not necessarily reproduce the structure of the filaments that are used as seeds[50]. Although structural templating may happen more readily in cultured cells, even there the biochemical environment seems to affect which structures form[51]. Moreover, observations that even tau monomers[52,53] can seed aggregation in cells are difficult to understand with the existing models of seeded aggregation. Similarly to the experiments described here, time-resolved cryo-EM of in vitro seeded assembly reactions may provide a better understanding of the molecular mechanisms of templated seeding.

Distinct structures of tau filaments extracted from the brains of individuals with different diseases have led to a structure-based classification of tauopathies[10]. Extra densities in the cryo-EM maps suggest

that unidentified molecules may co-assemble with tau. The biochemical environment in which filaments are formed may thus play a defining role in the formation of specific structures in the different diseases. Our in vitro assembly reactions recapitulate this for Alzheimer's disease and CTE. The presence of magnesium chloride or sodium chloride in the buffer defines the structure of the final reaction products, as well as those of intermediates that form after 180 min. However, independently of the reaction conditions, the same FIA forms after 120 min. If the FIA can form in a wide range of biochemical environments, then the same primary nucleation event is likely to lead to the formation of filaments from other tauopathies. Identification of the corresponding reaction conditions will be required to test this hypothesis.

Our experiments reveal the structures of intermediate amyloid species that form during the in vitro assembly of tau(297–391) into filaments with the Alzheimer or CTE fold. Whether these short-lived species can be isolated as biochemically stable entities against which antibodies could be raised remains to be tested. As small oligomeric species would be difficult to visualize using cryo-EM, we cannot exclude their presence in our experiments. Nevertheless, our data suggest that the in vitro formation of disease-specific tau folds happens through a mechanism whereby tau monomers nucleate directly into FIAs, which then grow and turn into mature filaments through multiple routes of secondary nucleation, involving many different intermediate amyloids. This model, in which prefibrillar oligomeric species are not required, provides a new perspective on the molecular mechanisms of amyloid formation, with important implications for the development of new therapies.

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

## Methods

### Expression and purification of tau(297–391)

Expression of tau(297–391) was carried out in *Escherichia coli* BL21 (DE3)-gold cells (Agilent Technologies), as described previously[54]. In brief, one plate of cells was resuspended in 1 l 2xTY (tryptone yeast) supplemented with 100 mg l$^{-1}$ ampicillin and grown to an optical density of 0.8 at 600 nm. For uniformly $^{15}$N- and $^{13}$C-labelled tau, bacteria were grown in isotope-enriched M9 minimal medium containing 1 g l$^{-1}$ of [$^{15}$N]ammonium chloride and 2 g l$^{-1}$ of [$^{13}$C]glucose (Sigma) supplemented with 1.7 g l$^{-1}$ yeast nitrogen base (Sigma). Cells were induced by the addition of 1 mM IPTG for 4 h at 37 °C, collected by centrifugation (4,000*g* for 20 min at 4 °C), resuspended in washing buffer (50 mM MES at pH 6.0; 10 mM EDTA; 10 mM dithiothreitol (DTT), supplemented with 0.1 mM phenylmethylsulfonyl fluoride and cOmplete EDTA-free protease cocktail inhibitors, at 10 ml per gram of pellet) and heated at 95 °C for 5 min. Cell lysis was carried out using sonication (at 40% amplitude using a Sonics VCX-750 Vibracell Ultra Sonic Processor for 7 min, 5 s on/10 s off). Lysed cells were centrifuged at 20,000*g* for 35 min at 4 °C, filtered through 0.45-µm cutoff filters and loaded onto a HiTrap CaptoS 5-ml column (GE Healthcare) for cation exchange. The column was washed with 10 column volumes of washing buffer and eluted using a gradient of washing buffer containing 0–1 M NaCl. Fractions of 3.5 ml were collected and analysed by SDS–polyacrylamide gel electrophoresis (PAGE; Tris-glycine 4–20% gels). Protein-containing fractions were pooled and precipitated using 0.28 g ml$^{-1}$ ammonium sulfate and left on a rocker for 30 min at 4 °C. Precipitated proteins were then centrifuged at 20,000*g* for 35 min at 4 °C, and resuspended in 2 ml of 10 mM phosphate buffer at pH 7.2 with 10 mM DTT, and loaded onto a 16/600 75-pg size-exclusion column. Size-exclusion fractions were analysed by SDS–PAGE, and protein-containing fractions were pooled and concentrated to 20 mg ml$^{-1}$ using molecular weight concentrators with a cutoff filter of 3 kDa. Purified protein samples were flash frozen in 50–100 µl aliquots for future use. Protein concentrations were determined using a NanoDrop2000 (Thermo Fisher Scientific).

### Analytical ultracentrifugation

Tau at a concentration of 6 mg ml$^{-1}$ in 10 mM sodium phosphate, pH 7.2, 10 mM DTT was loaded into 12-mm two-sector cells, placed in an An50Ti rotor and centrifuged at 50,000 r.p.m. at 4 °C using an Optima XL-I analytical ultracentrifuge (Beckman). The data were analysed in SEDFIT-16.1c[55] using a *c*(*S*) distribution model to determine the concentrations (*c*) of species with sedimentation coefficients (*S*) that fit the sedimentation profiles. The partial specific volumes ($\bar{v}$), density and viscosity of the buffer were calculated using Sednterp[56]. Data were plotted with the program GUSSI 1.4.2 (ref. 57) and Prism 9.5.1 (GraphPad Software).

### NMR

Solution-state NMR data were acquired at 278 K using 14.1-tesla (T), 18.8-T and 22.3-T Bruker Avance III spectrometers fitted with 5-mm TCI triple resonance cryoprobes, operating at proton frequencies of 600, 800 and 950 MHz, respectively. All NMR samples were prepared in 50 mM phosphate buffer at pH 7.4, with 150 mM NaCl, 10 mM DTT and 5% D$_2$O as a lock solvent.

Assignment of backbone NH, N, Cα, Cβ and C′ resonances of isotopically enriched ($^{15}$N and $^{13}$C) human tau(297–391) at 300 µM was completed at 600 MHz. Standard three-dimensional (3D) datasets were acquired as pairs to provide own and preceding carbon connectivities, using 18–39% non-uniform sampling to aid faster data acquisition. Both the HNCO and HN(CA)CO and the HNCA and HN(CO)CA experimental pairs were collected with 2,048, 64 and 128 complex points in the $^1$H, $^{15}$N and $^{13}$C dimensions, respectively. The CBCA(CO)NH and HNCACB pair were collected with 2,048, 64 and 96 complex points in the $^1$H, $^{15}$N and $^{13}$C dimensions, respectively. Additional $^{15}$N connectivities were established using (H)N(COCA)NNH experiments with 2,048, 80 and 128 complex points in the $^1$H, direct $^{15}$N and indirect $^{15}$N dimensions, respectively. C′-detect experiments c_hcacon_ia3d and c_hcanco_ia3d were also collected to aid backbone assignment with 1,024, 64 and 128 points collected in the direct $^{13}$C, direct $^{15}$N and indirect $^{13}$C dimensions, respectively.

All raw NMR data were processed using Topspin versions 3.2 or 4 (Bruker), or using NMRPipe[58], with compressed sensing for reconstruction of non-uniform sampling data (through qMDD)[59] and analysed using NMRFAM-Sparky and MARS[60].

Secondary chemical shift analysis was carried out to probe secondary structure elements. Random coil Cα, Cβ and C′ values for tau(297–391) were calculated[61–63], and subtracted from the experimentally derived values, to give ΔCα, ΔCβ and ΔC′, respectively. Negative values for (ΔCα + ΔC′)/2 − ΔCβ are indicative of residues in an extended backbone conformation; positive values indicate helical conformations.

Molecular motions were probed at proton frequencies of 600, 800 and 950 MHz ($^{15}$N frequencies of 60.8, 80.8 and 96.3 MHz, respectively). Longitudinal relaxation was probed with a standard Bruker $^{15}$N T1 pseudo-3D experiment, collected with 10-, 20-, 40-, 80-, 120-, 160-, 320-, 640-, 1,280- and 2,000-ms delays and a standard Bruker $^{15}$N T2 pseudo-3D experiment with 16.9-, 33.8-, 50.7-, 67.6-, 84.5-, 101.4-, 118.3-, 169-, 202.8- and 253.5-ms delays. Both experiments used a recovery delay of 5 s. Picosecond motions were monitored with the standard Bruker interleaved 2D $^{15}$N{$^1$H} heteronuclear nuclear Overhauser effect experiment with a recovery delay of 5 s. Additional longitudinal and transverse cross-correlated cross-relaxation measurements were collected as described previously[64], with Δ*T* relaxation periods of 100 and 40 ms, respectively. Calculation of the exchange-free R$_2$ rates ($R_{2eff}$) was carried out as described previously[65].

IMPACT analysis was completed using a Mathematica (Wolfram) script, as described previously[25]. Corresponding longitudinal and transverse cross-correlated cross-relaxation measurements and $^{15}$N{$^1$H} heteronuclear nuclear Overhauser effect values collected at three different field strengths were used to create a spectral density analysis landscape. These frequency-specific data were fitted to multiple correlation times, varying in number from 4 to 9, and across a range of correlation times (from 2 ps–2 ns up to 100 ps–100 ns) using Monte Carlo simulations. The Akaike information criterion was used to evaluate the statistical relevance of each condition, indicating that the best representation comprised five times scales, ranging from 36 ps to 36 ns.

### Assembly of tau

Filaments were assembled as described previously[50], with minor modifications. Assembly reactions were carried out in aliquots of 40 µl of purified monomeric tau(297–391) at 6 mg ml$^{-1}$, in a 384-well microplate that was sealed and placed in a Fluostar Omega (BMG Labtech). PHF reactions were carried out in 10 mM phosphate buffer at pH 7.2, 100 mM MgCl$_2$ and 10 mM DTT. Previously, we reported the in vitro assembly of tau(297–391) into PHFs with 200 mM MgCl$_2$ (ref. 3). The different concentrations of MgCl$_2$ did not affect the formation of PHFs. CTE reactions were carried out in 50 mM phosphate buffer at pH 7.2, 150 mM NaCl and 10 mM DTT. Reactions were carried out for 720 min using 200 r.p.m. orbital shaking at 37 °C. Our previous assembly reactions[3] were carried out over 48 h. Waiting longer increases the total amount of PHFs or CTE filaments, but also leads to a clumping together of filaments, which complicates cryo-EM analysis. For continuous ThT monitoring, 1.5 µM of ThT was added to the reaction and measurements were taken every 10 min. For offline monitoring, multiple replicates of the reactions were carried out, and for each time point, ThT was added to the entire volume of a separate reaction replicate at a concentration of 1.5 µM.

### Quantification of pelletable tau

Multiple replicas of the reactions were also carried out for the quantification of pelletable tau. At 0, 60, 90, 120, 360 and 720 min, the

entire volume of individual reaction replicas was collected for ultra-centrifugation. Reactions were centrifuged at 400,000*g* at 20 °C for 15 min in polycarbonate centrifuge tubes (Beckman Coulter). The pellets were resuspended in 40 µl reaction buffer, to match the volume of the supernatants. Loading buffer was added to supernatants and pellets, which were then heated for 5 min at 95 °C, and 1.5 µl of each was run by SDS–PAGE (4–20% Tris-glycine gels). Band intensities were quantified using ImageJ and data were plotted using Prism 9.5.1 (GraphPad Software).

### Cryo-EM data acquisition

At specific time points, the microplates were taken from the shaker and 3 µl of the reaction mixture were applied to glow-discharged R1.2/1.3, 300 mesh carbon Au grids. The grids were plunge-frozen in liquid ethane using a Vitrobot Mark IV (Thermo Fisher Scientific). After taking each aliquot, the microplate was resealed and returned to the shaker to continue the assembly reaction within 10 min.

Cryo-EM data were acquired at the Medical Research Council (MRC) Laboratory of Molecular Biology (LMB) and at the Research and Development facility of Thermo Fisher Scientific in Eindhoven (TFS). At LMB, images were recorded on a Krios G2 (Thermo Fisher Scientific) electron microscope that was equipped with a Falcon-4 camera (Thermo Fisher Scientific) without an energy filter. At TFS, images were recorded on a Krios G4 (Thermo Fisher Scientific) with a cold field-emission gun, a Falcon-4 camera and a Selectris X (Thermo Fisher Scientific) energy filter that was used with a slit width of 10 eV. All images were recorded at a dose of 30–40 electrons per square ångström using EPU software (Thermo Fisher Scientific) and converted to tiff format using relion_convert_to_tiff[66] before processing.

### Cryo-EM data processing

Video frames were gain corrected, aligned and dose weighted using RELION's motion correction program[67]. Contrast transfer function (CTF) parameters were estimated using CTFFIND-4.1 (ref. 68). Helical reconstructions were carried out using RELION-4.0 (refs. 29,69). Filaments were picked manually or automatically using a modified version of Topaz[70,71]. Picked particles were extracted in boxes of either 1,024 or 768 pixels and downscaled to 256 or 128 pixels for initial classification.

Reference-free 2D classification, with at least 150 classes and ignoring the CTF until its first peak, was carried out for at least 35 iterations to assess the presence of different polymorphs and crossover distances. Polymorphs were identified by a new hierarchical clustering approach that was inspired by the CHEP algorithm[72] (see below). Selected particles were re-extracted in boxes of 384 pixels for initial 3D refinement. Initial 3D references were generated de novo from 2D class average images using relion_helix_inimodel2d[73]. For the FIA and structures that had low particle numbers (<5,000), a new algorithm using regularization by denoising[74] improved initial refinements, as conventional refinements resulted in high noise levels in the reconstruction due to overfitting. Subsequently, 3D classifications and 3D auto-refinements were used to select particles leading to the best reconstructions and to optimize helical parameters. For some datasets, 3D classification was also used to separate out closely related polymorphs. For maps that were used for atomic modelling, Bayesian polishing[67] and CTF refinement[75] were used to increase resolution. Final maps were sharpened using standard post-processing procedures in RELION, and reported resolutions were estimated using a threshold of 0.143 in the Fourier shell correlation (FSC) between two independently refined half-maps (Supplementary Figs. 49–52). The handedness of cryo-EM maps with resolutions beyond 2.9 Å was determined from the presence of densities for main-chain carbonyl oxygens. For all other maps, the handedness was determined on the basis of substructures that were also present in maps that were solved at resolutions beyond 2.9 Å.

### Polymorph identification and quantification

Picked filaments were hierarchically clustered by the unweighted pair group method with arithmetic mean, on the basis of the cosine distance of the 2D class assignment distributions of particles for each filament from an initial 2D classification. Clusters were selected either by flattening the dendrogram at a specified threshold or interactively. Clusters below a minimum threshold of particles, typically 1,000, were merged. Additional 2D classifications were carried out for each identified cluster, iterating the clustering and 2D classification procedure until visually homogeneous populations of 2D classes were obtained. Filamentous class averages were then selected and output particles were used for refinement.

Reported percentages of filaments in each dataset were calculated on the basis of the number of extracted particles used for the initial refinement of a particular filament type, relative to the total number of picked particles. For auto-picked datasets, an initial round of reference-free 2D classification was sometimes used to remove false positives from the picking procedure first. The reported percentages may not reflect the relative amounts of filament types in the original assembly reactions because of limitations in our image analysis and because some filament types may disperse better than others in the grid holes.

### Atomic modelling

Atomic models were built either manually using COOT[76] or automatically using ModelAngelo[77]. Coordinate refinement of models comprising three β-rungs was carried out in ISOLDE[78]. To ensure consistency, dihedral angles from the middle rung were applied to the top and bottom rungs. Subsequently, separate model refinements were carried out on the first half-map for each refined structure. The resulting models were then evaluated by comparing them to this half-map (FSC$_{work}$), as well as to the other half-map (FSC$_{test}$) to monitor overfitting (Supplementary Figs. 49–52). Figures of structures, including electrostatic potential and hydrophobicity surfaces, were prepared using ChimeraX[79]. Extended Data Fig. 8 was prepared using the Amyloid Illustrator software[80].

### Reporting summary

Further information on research design is available in the Nature Portfolio Reporting Summary linked to this article.

### Data availability

Cryo-EM maps of all unique structures have been deposited in the Electron Microscopy Data Bank (EMDB). Refined atomic models in those maps with resolutions sufficient for atomic modelling have been deposited in the Protein Data Bank (PDB). The accession numbers (which are also listed in Extended Data Fig. 7) are as follows—FIA: EMDB EMD-17806, PDB 8PPO; AD-MIA1: EMDB EMD-18070, PDB 8Q27; AD-MIA2: EMDB EMD-18109, PDB 8Q2J; AD-MIA3: EMDB EMD-18111, PDB 8Q2K; AD-MIA4: EMDB EMD-18112, PDB 8Q2L; AD-MIA5: EMDB EMD-18215, PDB 8Q7F; AD-MIA6: EMDB EMD-18219, PDB 8Q7L; AD-MIA7: EMDB EMD-18224, PDB 8Q7M; AD-MIA8: EMDB EMD-18228, PDB 8Q7P; AD-MIA9: EMDB EMD-18348; AD-MIA10: EMDB EMD-18250, PDB 8Q8C; AD-MIA11: EMDB EMD-18233, PDB 8Q7T; AD-LIA1: EMDB EMD-18344; AD-LIA2: EMDB EMD-18249, PDB 8Q88; AD-LIA3: EMDB EMD-18347; AD-LIA4: EMDB EMD-18252, PDB 8Q8E; AD-LIA5: EMDB EMD-18253, PDB 8Q8F; AD-LIA6: EMDB EMD-18251, PDB 8Q8D; AD-LIA7: EMDB EMD-18254, PDB 8Q8L; AD-LIA8: EMDB EMD-18331, PDB 8QCP; AD-THF: EMDB EMD-18259, PDB 8Q8S; AD-QHF: EMDB EMD-18349; AD-PHFb: EMDB EMD-18255, PDB 8Q8M; AD-PHFa: EMDB EMD-18258, PDB 8Q8R; CTE-MIA1: EMDB EMD-18261, PDB 8Q8U; CTE-MIA2: EMDB EMD-18354; CTE-MIA3: EMDB EMD-18262, PDB 8Q8V; CTE-MIA4: EMDB EMD-18263, PDB 8Q8W; CTE-MIA5: EMDB EMD-18264, PDB 8Q8X; CTE-MIA6: EMDB EMD-18265, PDB 8Q8Y; CTE-MIA7: EMDB EMD-18266, PDB 8Q8Z;

CTE-MIA8: EMDB EMD-18271, PDB 8Q98; CTE-MIA9: EMDB EMD-18270, PDB 8Q97; CTE-MIA10: EMDB EMD-18272, PDB 8Q99; CTE-MIA11: EMDB EMD-18273, PDB 8Q9A; CTE-MIA12: EMDB EMD-18333, PDB 8QCR; CTE-MIA13: EMDB EMD-18275, PDB 8Q9B; CTE-MIA14: EMDB EMD-18276, PDB 8Q9C; CTE-MIA15: EMDB EMD-18277, PDB 8Q9D; CTE-MIA16: EMDB EMD-18355; CTE-MIA17: EMDB EMD-18228, PDB 8Q7P; CTE-MIA18: EMDB EMD-18278, PDB 8Q9E; CTE-MIA19: EMDB EMD-18356; CTE-MIA20: EMDB EMD-18357; CTE-LIA1: EMDB EMD-18358; CTE-LIA2: EMDB EMD-18359; CTE-LIA3: EMDB EMD-18279, PDB 8Q9F; CTE-LIA4: EMDB EMD-18281, PDB 8Q9H; CTE-LIA5: EMDB EMD-18280, PDB 8Q9G; CTE-LIA6: EMDB EMD-18282, PDB 8Q9I; CTE-LIA7: EMDB EMD-18283, PDB 8Q9J; CTE-LIA8: EMDB EMD-18360; CTE-LIA9: EMDB EMD-18361; CTE-LIA10: EMDB EMD-18362; CTE-LIA11: EMDB EMD-18363; CTE-LIA12: EMDB EMD-18364; CTE-LIA13: EMDB EMD-18284, PDB 8Q9K; CTE-LIA14: EMDB EMD-18285, PDB 8Q9L; CTE-LIA15: EMDB EMD-18365; CTE-LIA16: EMDB EMD-18366; CTE-LIA17: EMDB EMD-18287, PDB 8Q9O; CTE type I: EMDB EMD-18286, PDB 8Q9M; CTE type II: EMDB EMD-18448, PDB 8QJJ.

## Code availability

The RELION software is available at https://github.com/3dem/relion. Scripts for clustering filament types, as well as for generating dataset summaries as shown in Supplementary Figs. 1–48, are available at https://github.com/dbli2000/FilamentTools and have been incorporated into version 5.0 of the RELION software.

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

**Acknowledgements** We thank C. Charlier and F. Ferrage for providing the Mathematica script for IMPACT analysis; H. Wang and T. P. J. Knowles for helpful discussions; J. Grimmett, T. Darling and I. Clayson for help with high-performance computing; and M. Wilkinson and R. A. Crowther for critical reading of the manuscript. This work was supported by the facilities for biophysics, electron microscopy, NMR and scientific computing of the MRC Laboratory of Molecular Biology, and by the Francis Crick Institute through provision of access to the MRC Biomedical NMR Centre. The Francis Crick Institute receives its core funding from Cancer Research UK (CC1078), the UK MRC (CC1078) and the Wellcome Trust (CC1078). This work was supported by the MRC, as part of United Kingdom Research and Innovation (MC_U105184291 to M.G. and MC_UP_A025-1013 to S.H.W.S.), and a Marshall scholarship to D.L.

**Author contributions** S.L. carried biochemistry and filament assembly; S.L. and A.K. acquired cryo-EM data; S.L., D.L., D.K., A.G.M. and S.H.W.S. analysed cryo-EM data; D.L. and D.K. wrote software; J.L.W. and S.M.V.F. carried out NMR; S.H.M. carried out analytical ultracentrifugation; M.G. and S.H.W.S. supervised the project. All authors contributed to the writing of the manuscript.

**Competing interests** The authors declare no competing interests.

**Additional information**
**Correspondence and requests for materials** should be addressed to Michel Goedert or Sjors H. W. Scheres.

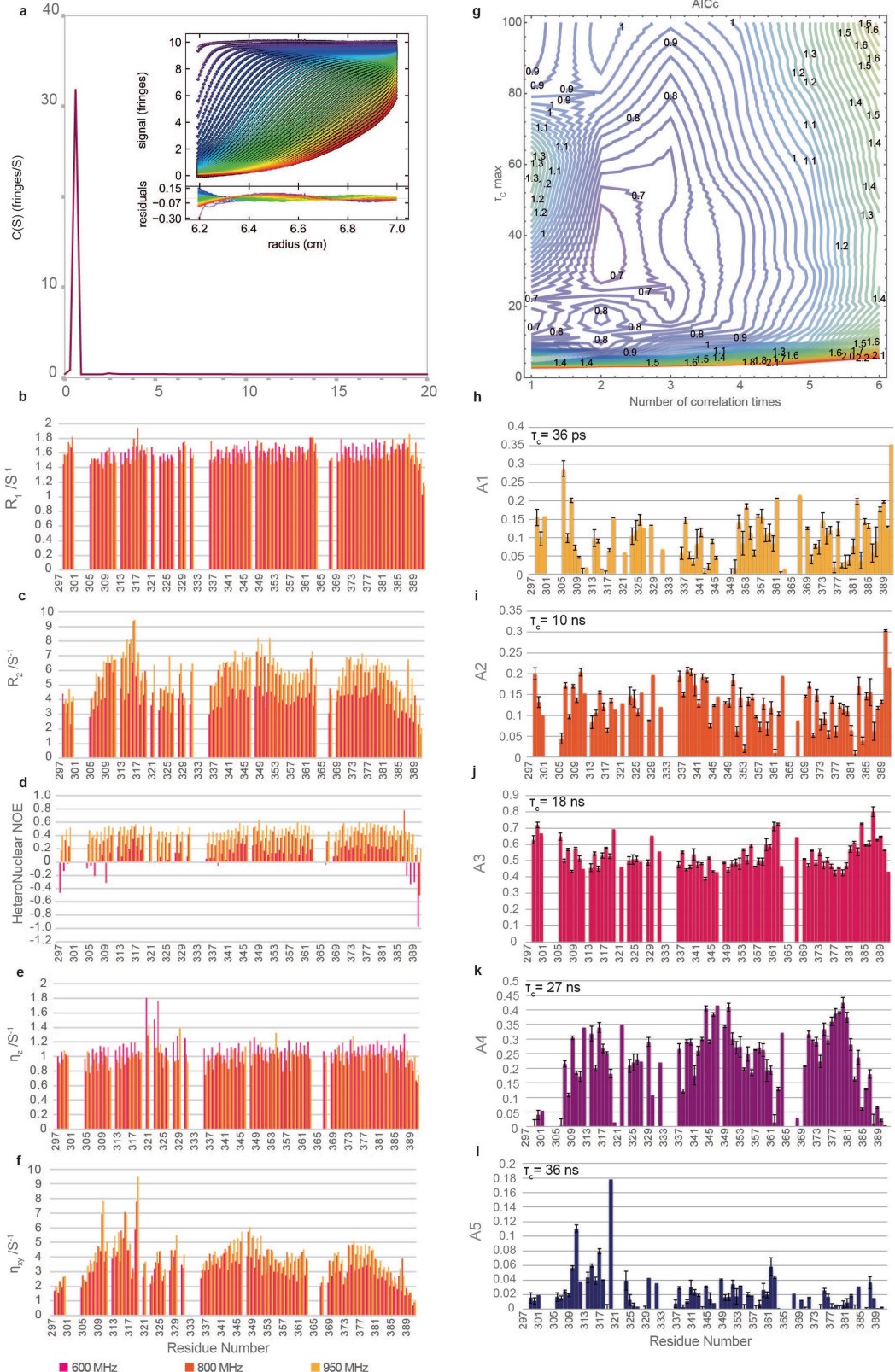

**Extended Data Fig. 1** | See next page for caption.

**Extended Data Fig. 1 | Analytical ultracentrifugation and nuclear magnetic resonance. a**. Analytical ultracentrifugation (AUC) sedimentation velocity analysis of tau(297–391) in solution. The c(S) distribution shows tau(297–391) sedimented with coefficient of 0.6 S ($S_{w,20}$ = 1.0 S) with a frictional ratio of 1.777 corresponding to a mass of 10.3 kDa, consistent with a monomer in an extended conformation. The panel inset shows interference profiles with best fits of a c(S) model (coloured lines) and their residuals to the fits underneath. The different colours represent scans at different times: blue is the earliest time points where very little material has sedimented; through to red where the material has sedimented. **b-d**. Longitudinal $R_1$, transverse $R_2$ and heteronuclear $^{15}$N{$^1$H} NOE measurements of tau(297–391) collected at 600 (magenta), 800 (orange) and 950 MHz (yellow) proton resonance frequencies. **e-f**. Exchange-free longitudinal and transverse cross-relaxation experiments collected at the same three proton resonance frequencies. **g**. The Akaike information criterion for assessment of the fitness of a range of correlation times (between 4–9) and time scale conditions $\tau_{min} - \tau_{max}$ (ranging from 2ps-2ns to 100ps-100ns) that best fit the spectra density analysis. **h-l**. Individual distributions of five coefficients ($A_1$-$A_5$) of the five correlation times $\tau_c$ = 36 ps, 10 ns, 18 ns, 27 ns and 36 ns as determined by the IMPACT analysis. Bars represent the mean of 11 Monte Carlo fits, with error bars showing the standard deviation.

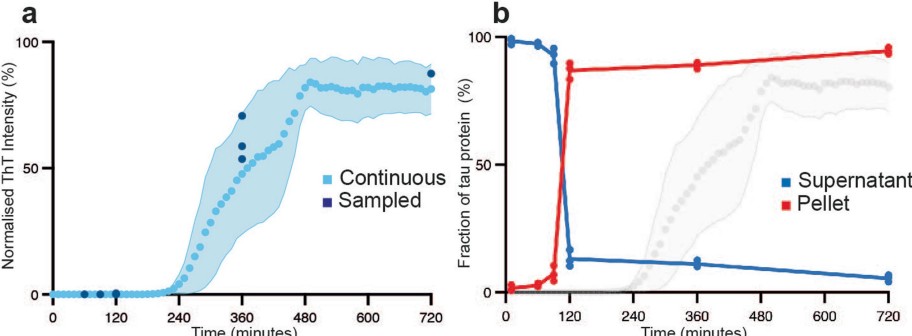

**Extended Data Fig. 2 | ThT fluorescence and pelletable tau in the CTE reactions. a**. ThT fluorescence profile of the CTE reactions. Light blue circles indicate the average of 3 replicates of continuous ThT monitoring; light blue shading indicates the standard deviation among 3 replicates; dark blue circles represent off-line ThT measurements. **b**. The amount of tau in the pellet and in the supernatant (in % of the total amount of tau) after centrifugation for 15 min at 400,000 g, as quantified by SDS-PAGE.

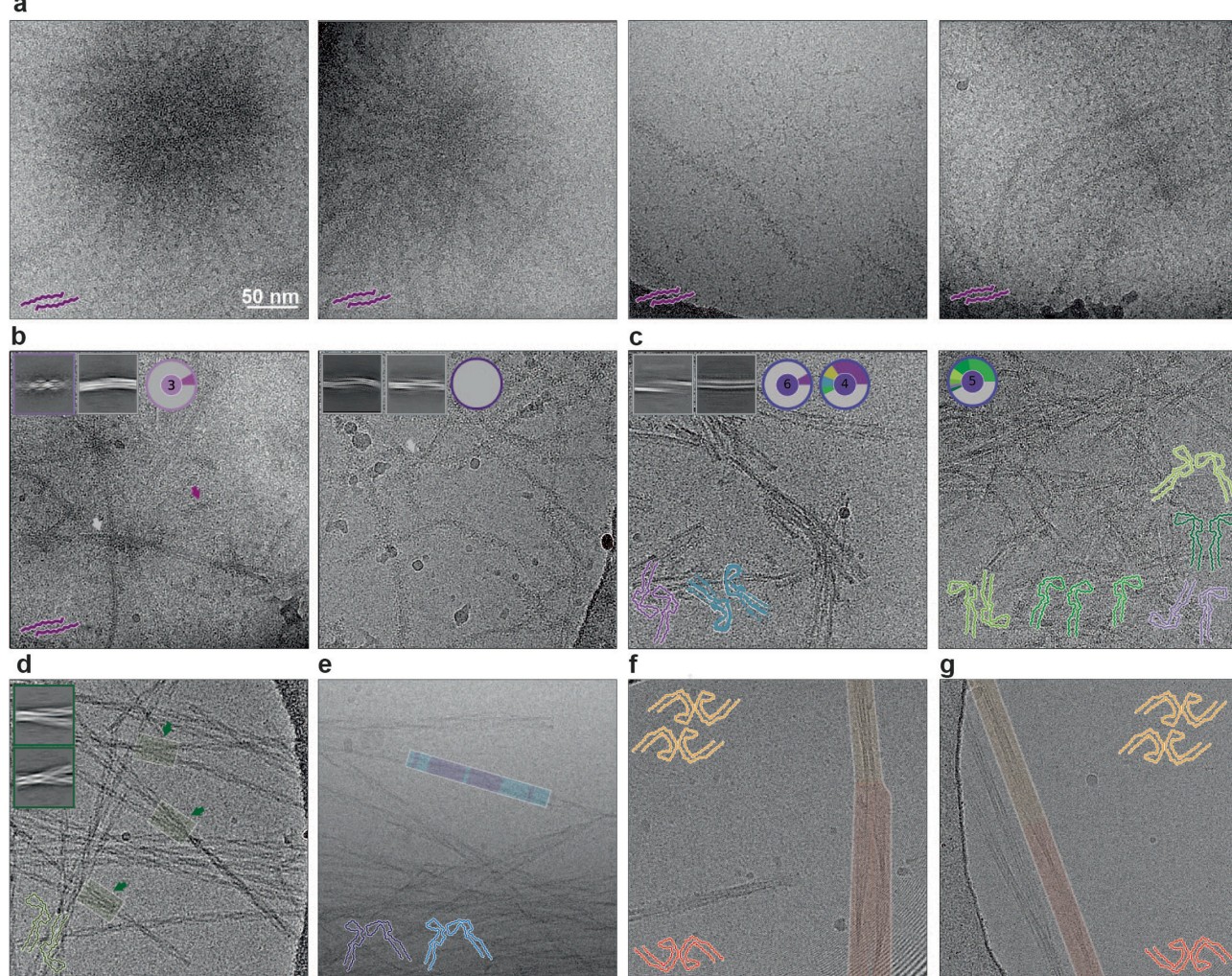

**Extended Data Fig. 3 | Various observations in micrographs. a.** At 120 min, micrographs of the PHF reactions showed FIAs (cartoon at bottom left). In some instances, many FIAs appeared to originate from a single point, reminiscent of nucleation and growth of a crystal from an impurity (left two micrographs). In other instances, longer isolated FIAs were observed. **b.** At 140 min in the PHF reactions, some FIAs remained but most filaments yielded images that did not allow 3D reconstruction. Insets show 2D class averages; circular pie charts show the distribution of unsolvable filaments (grey) versus structures solved (coloured). **c.** At 160 min in the CTE reactions, nine structures could be solved from three replicates. **d.** At 180 min in the CTE reactions, some filament types appear to be branching, with specific structures in 2D class averages (insets). **e.** At 180 min in the CTE reactions, some filaments consist of multiple filament types (purple or blue). **f.** At 360 min in the PHF reactions, some QHFs (yellow) appear to be branching into two separate PHFs (red). **g.** At 360 min in the PHF reactions, some QHFs appear to convert into PHFs. Scale bar of 50 nm applies to all micrographs.

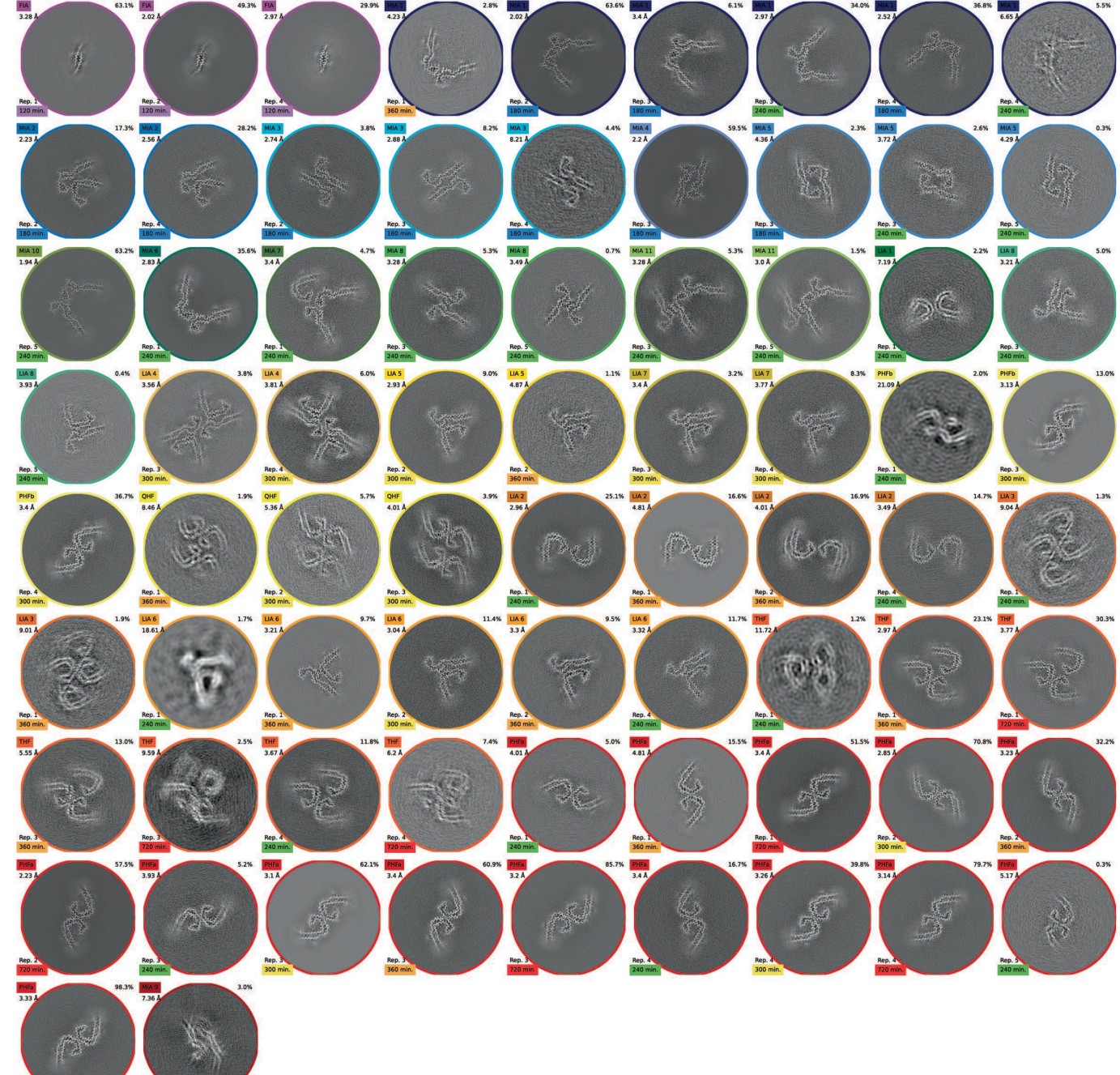

**Extended Data Fig. 4 | Cryo-EM reconstructions from the PHF reactions.** Projected slices, with an approximate thickness of 4.7 Å, orthogonal to the helical axis for the filaments formed in the PHF reactions. Filament names and resolutions are indicated in the top left; percentages of filament types in each cryo-EM data set are shown in the top right and the replicate and time point are indicated in the bottom left of each image. Circles around the slices are coloured as the structures of Fig. 4 in the main text.

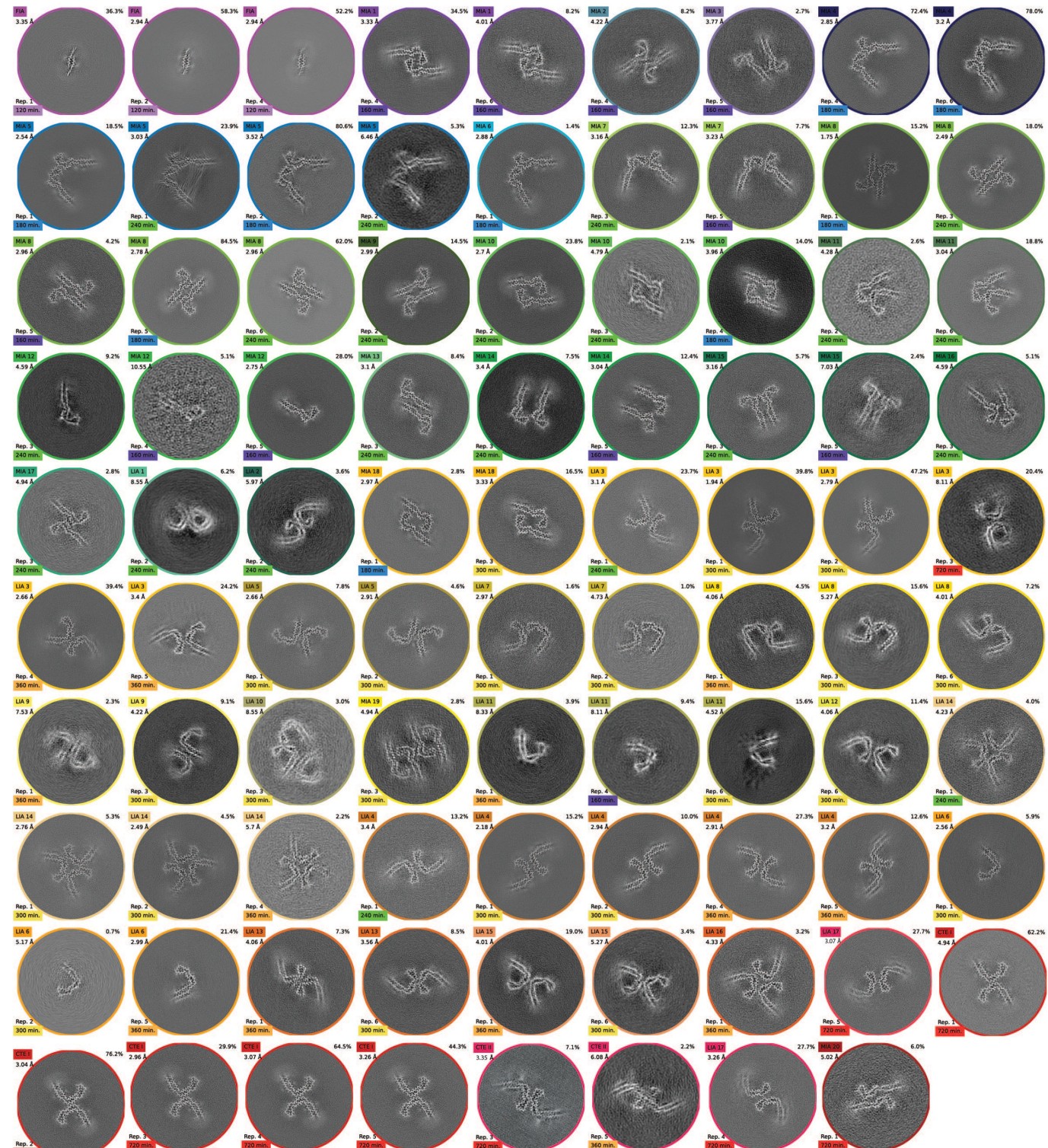

**Extended Data Fig. 5 | Cryo-EM reconstructions from the CTE reactions.** Projected slices, with an approximate thickness of 4.7 Å, orthogonal to the helical axis for the filaments formed in the CTE reactions. Filament names and resolutions are indicated in the top left; percentages of filament types in each cryo-EM data set are shown in the top right and the replicate and time point are indicated in the bottom left of each image. Circles around the slices are coloured as the structures of Fig. 4 in the main text.

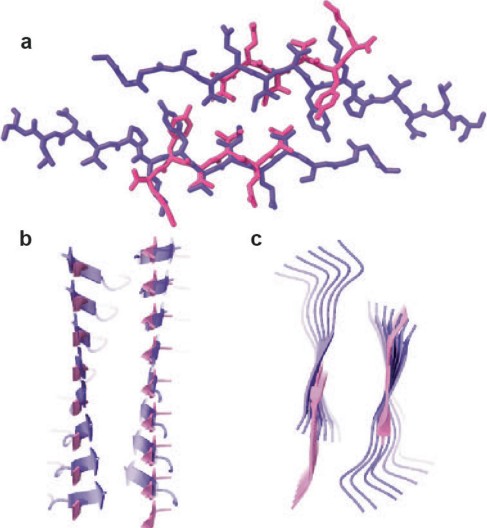

**Extended Data Fig. 6 | Twisted β-sheets in the FIA and other structures.**
**a**. Top view of an all-atom representation of the FIA (purple) and one of the crystal structures of the $_{306}$VQIVYK$_{311}$ peptide (pink; PDB entry 2ON9[30]) illustrate similarity in their packing interface. **b**. Side view of the crystal structure and the FIA. β-Sheets in the FIA are twisted; β-sheets in the crystal structure are straight. **c**. As in panel **b**, but top view.

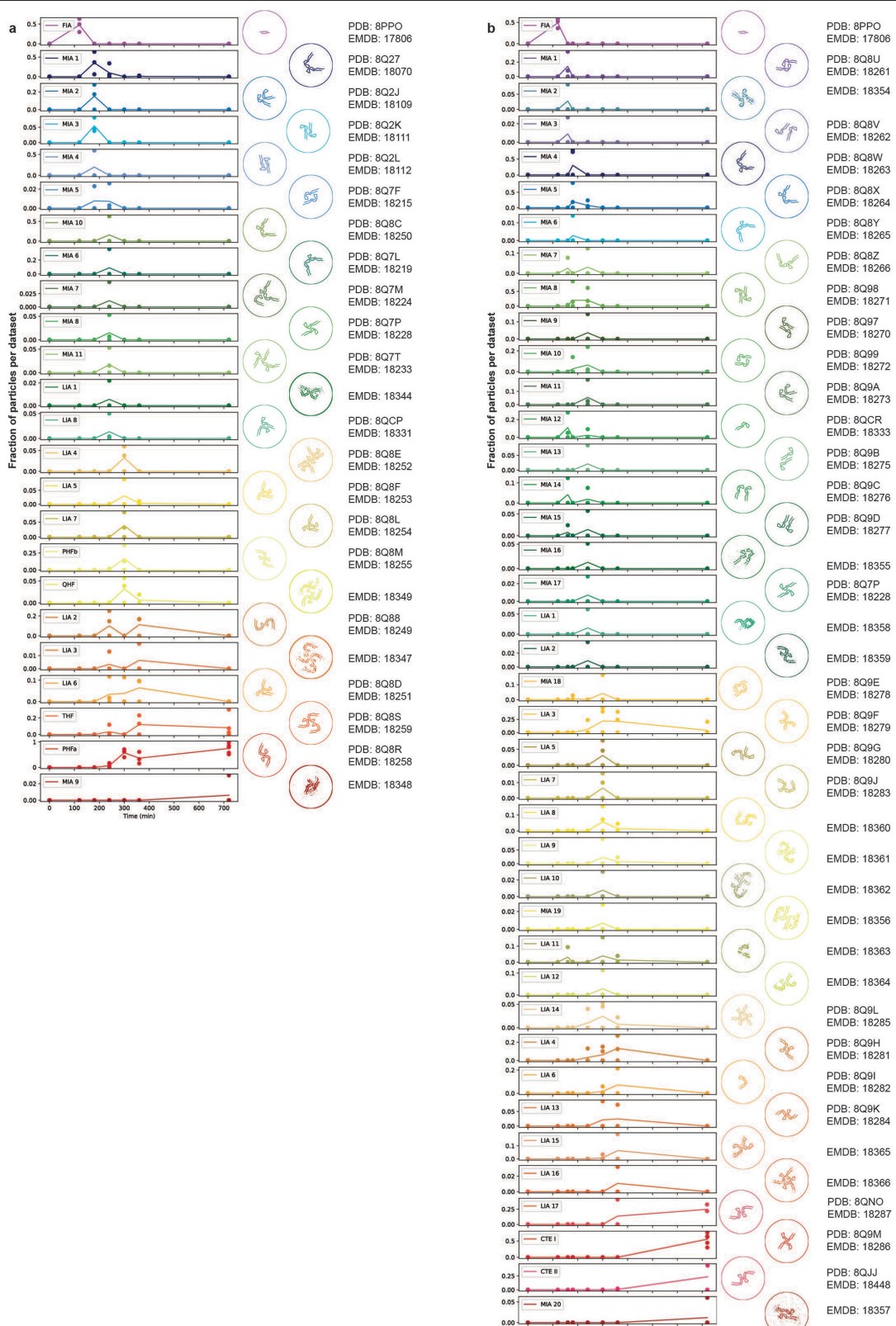

**Extended Data Fig. 7 | Time-dependent abundance of unique structures.**
**a**. Relative abundance, as computed from particle distributions, for each unique filament structure in the PHF reactions is shown for the different time points. Circles indicate relative abundance for individual replicates; lines indicate the average over multiple replicates. For structures of sufficient resolution to allow atomic modelling, PDB and EMDB entry codes are also shown. For other maps, only EMDB entry codes are shown. **b**. As in **a**, but for the CTE reactions.

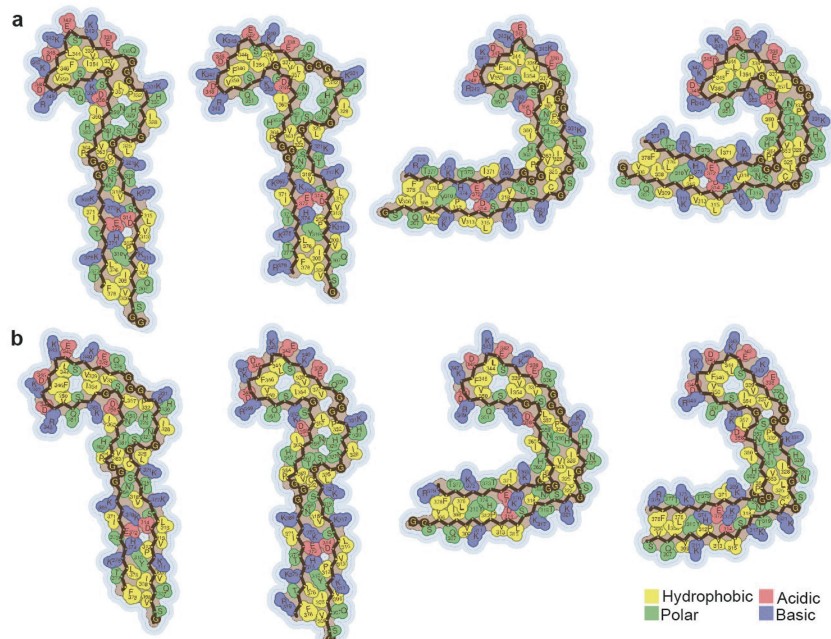

**Extended Data Fig. 8 | Schematic representations of J- and C-shaped protofilaments. a**. Schematic representation as defined in the Amyloid Illustrator[80], of the residue packing of J- and C-shaped protofilaments from the PHF reactions, as shown in Fig. 5, illustrates the presence of voids near the $_{332}$PGGG$_{335}$ motif in J-shaped protofilaments from the PHF reaction. **b**. As in **a**, but for protofilaments from the CTE reactions.

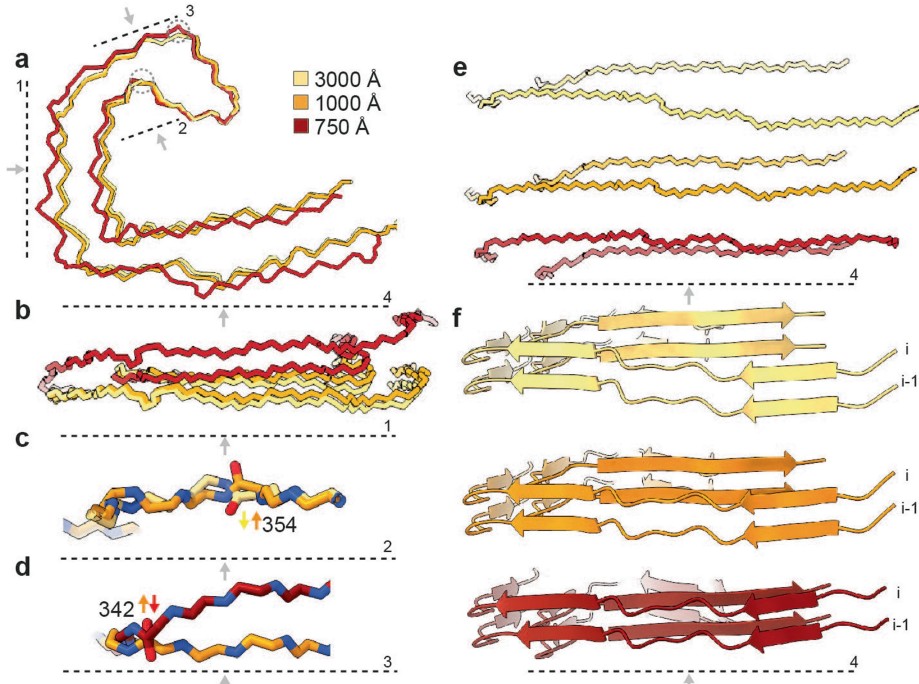

**Extended Data Fig. 9 | PHF crossover distances. a**. Backbone traces, aligned at amino acids 339–354, for PHFs with crossover distances of 750 Å (red), 1000 Å (orange) and 3000 Å (yellow). Grey dotted lines and arrows indicate viewing planes and directions in subsequent panels. **b**. Side view of amino acids 305–320 and 365–380. **c**. The carbonyl of isoleucine 354 flips from PHFs with a crossover distance of 1000 Å, compared to those with a crossover distance of 3000 Å. **d**. The carbonyl of glutamic acid 342 flips from PHFs with a crossover distance of 750 Å, compared to those with a crossover distance of 1000 Å. **e**. Side view of backbone traces of amino acids 305–320 and 365–380 for PHFs with crossover distances of 3000 Å (top), 1000 Å (middle) and 750 Å (bottom). **f**. As in panel **e**, but for cartoon representations of two β-rungs.

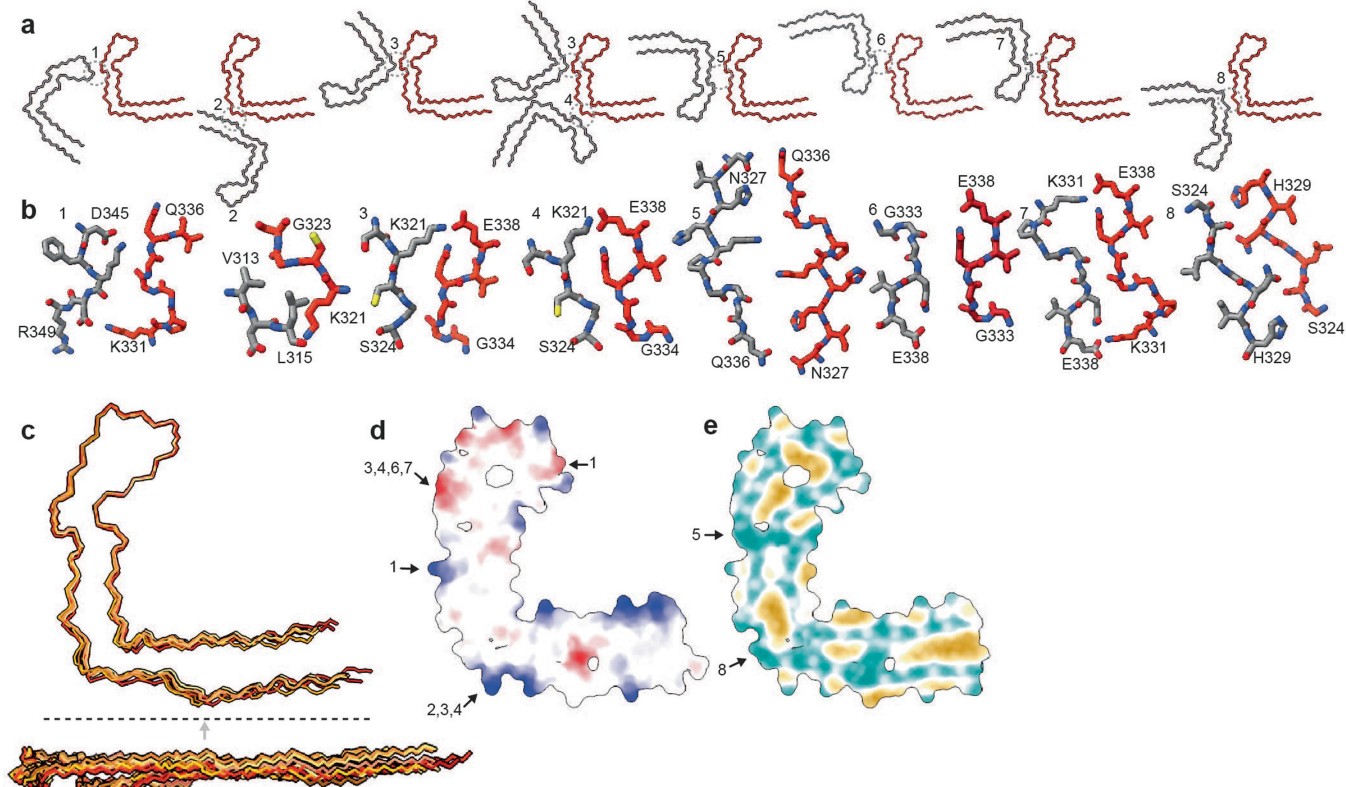

**Extended Data Fig. 10 | CTE protofilament interactions. a**. Backbone traces for C-shaped structures that are present at 300–720 min in the CTE reaction. The red protofilament is aligned across all structures; the grey protofilaments adopt varying orientations relative to the red one. The dashed circles and numbers are referred to in panels b, d and e. **b**. All-atom representation of the protofilament interactions shown in panel a. **c**. A superposition of backbone traces (top) and side view (bottom) shows that the different protofilaments from panel a adopt similar conformations. **d**. Coulomb electrostatic potential (positive charges in blue; negative charges in red) of the CTE protofilament. **e**. Hydrophobicity representation (hydrophobic parts in yellow; charged parts in cyan).

# Reporting Summary

## Statistics

For all statistical analyses, confirm that the following items are present in the figure legend, table legend, main text, or Methods section.

| n/a | Confirmed | |
|---|---|---|
| ☐ | ☒ | The exact sample size ($n$) for each experimental group/condition, given as a discrete number and unit of measurement |
| ☐ | ☒ | A statement on whether measurements were taken from distinct samples or whether the same sample was measured repeatedly |
| ☒ | ☐ | The statistical test(s) used AND whether they are one- or two-sided<br>*Only common tests should be described solely by name; describe more complex techniques in the Methods section.* |
| ☒ | ☐ | A description of all covariates tested |
| ☒ | ☐ | A description of any assumptions or corrections, such as tests of normality and adjustment for multiple comparisons |
| ☒ | ☐ | A full description of the statistical parameters including central tendency (e.g. means) or other basic estimates (e.g. regression coefficient) AND variation (e.g. standard deviation) or associated estimates of uncertainty (e.g. confidence intervals) |
| ☒ | ☐ | For null hypothesis testing, the test statistic (e.g. $F$, $t$, $r$) with confidence intervals, effect sizes, degrees of freedom and $P$ value noted<br>*Give P values as exact values whenever suitable.* |
| ☒ | ☐ | For Bayesian analysis, information on the choice of priors and Markov chain Monte Carlo settings |
| ☒ | ☐ | For hierarchical and complex designs, identification of the appropriate level for tests and full reporting of outcomes |
| ☒ | ☐ | Estimates of effect sizes (e.g. Cohen's $d$, Pearson's $r$), indicating how they were calculated |

*Our web collection on statistics for biologists contains articles on many of the points above.*

## Software and code

Policy information about availability of computer code

| Data collection | For cryo-EM: EPU (v2.3.079) (Thermofisher). For NMR: Topspin (v3.2 and 4) (Bruker), |
|---|---|
| Data analysis | RELION (v4 and v5), CTFFIND (v4.1), COOT (v0.9.8.7), ChimeraX (v1.6.1), ImageJ (v2.1.0/1.53c), Prism (v9), GUSSI (v1.4.2), Sednterp (v3.0.4), SEDFIT-16.1c, Topspin (v3.2, 3.6.3 and 4.0) ), NMRpipe (v10.9), qMDD (v3.1), NMRFAM-Sparky (v1.470) , MARS (v1.2), Mathematics (v13.1), FiTSuite (v0.1). In addition, we developed and used cryoEM image processing scripts in python called FilamentTools, which can be downloaded from https://github.com/dbli2000/FilamentTools. It has no version number. |

For manuscripts utilizing custom algorithms or software that are central to the research but not yet described in published literature, software must be made available to editors and reviewers. We strongly encourage code deposition in a community repository (e.g. GitHub). See the Nature Portfolio guidelines for submitting code & software for further information.

## Data

Policy information about availability of data

All manuscripts must include a data availability statement. This statement should provide the following information, where applicable:
- Accession codes, unique identifiers, or web links for publicly available datasets
- A description of any restrictions on data availability
- For clinical datasets or third party data, please ensure that the statement adheres to our policy

All structures described have been submitted to the EMDB and PDB. Accession codes have been listed.

# Research involving human participants, their data, or biological material

Policy information about studies with human participants or human data. See also policy information about sex, gender (identity/presentation), and sexual orientation and race, ethnicity and racism.

| Reporting on sex and gender | NA |
|---|---|
| Reporting on race, ethnicity, or other socially relevant groupings | NA |
| Population characteristics | NA |
| Recruitment | NA |
| Ethics oversight | NA |

Note that full information on the approval of the study protocol must also be provided in the manuscript.

# Field-specific reporting

Please select the one below that is the best fit for your research. If you are not sure, read the appropriate sections before making your selection.

☒ Life sciences  ☐ Behavioural & social sciences  ☐ Ecological, evolutionary & environmental sciences

For a reference copy of the document with all sections, see nature.com/documents/nr-reporting-summary-flat.pdf

# Life sciences study design

All studies must disclose on these points even when the disclosure is negative.

| Sample size | Six samples were made for the CTE reaction and five for AD reactions which were analysed by cryo-EM. Aliquots were taken at different time intervals to sample the assembly reaction (which is shown in the supplementary information and in Figure 4 in the main text). These sample sizes were chosen such that at least 3 samples were available for cryo-EM at each time point (as not enough volume is present in each sample to take aliquots for all time points and not all cryo-EM grids have suitable ice for imaging). Three samples from the CTE reaction and three samples from the AD reactions were used for continuous ThT monitoring to determine the kinetics of amyloid assembly in the reaction. Three samples of each condition were used to determine the amount of pelletable tau which was analysed by SDS-PAGE and shown in the supplementary information. These samples were also used for the off-line monitoring of ThT fluorescence. This was to test whether ThT molecule affected the kinetics of the assembly reaction which is discussed in the main text of the paper. |
|---|---|
| Data exclusions | No data has been excluded. An overview of all data is provided in the Supplementary Information file. |
| Replication | Experiments were performed 5 and 6 times to establish the pathway of intermediate folding, collecting data points at different time points. Not all cryo-EM grids contained suitable ice for imaging and there was not enough volume in them to sample all time points, requiring five or six replicate reactions to achieve at least 3 cryo-EM data sets for all time points for both reactions. For continuous kinetic studies (ThT measuring), experiments were performed three times. For pelletable material, experiments were performed three times. |
| Randomization | Because there is no assignment of data points to distinct groups, randomization was not applicable to this study. Perhaps the only relevant randomization is that of determining random half-sets of particles for resolution assessment in cryo-EM reconstructions. This randomization was done using random number generators inside the RELION program. |
| Blinding | No blinding was performed, as the risk for bias by the experimentalist was deemed irrelevant for this study. |

# Reporting for specific materials, systems and methods

We require information from authors about some types of materials, experimental systems and methods used in many studies. Here, indicate whether each material, system or method listed is relevant to your study. If you are not sure if a list item applies to your research, read the appropriate section before selecting a response.

## Materials & experimental systems

| n/a | Involved in the study |
|---|---|
| ☒ | Antibodies |
| ☒ | Eukaryotic cell lines |
| ☒ | Palaeontology and archaeology |
| ☒ | Animals and other organisms |
| ☒ | Clinical data |
| ☒ | Dual use research of concern |
| ☒ | Plants |

## Methods

| n/a | Involved in the study |
|---|---|
| ☒ | ChIP-seq |
| ☒ | Flow cytometry |
| ☒ | MRI-based neuroimaging |

