## [Peer Review File · Nature]

Manuscript Title: Disease-specific tau filaments assemble via polymorphic intermediates

Reviewer Comments & Author Rebuttals

Reviewer Reports on the Initial Version:

Referees' comments:

Referee #1 (Remarks to the Author):

This paper on assembly kinetics of amyloid tau filaments is a tour-de-force and shows how cryo-EM may be used to capture multiple intermediate states on the pathway to mature amyloid filaments. The novel results are very likely to have a broad impact on the understanding of tau neuropathologies and will likely shape a great deal of future work in this area. I had only a number of minor points to raise:

p. 2) It is argued from using AUC (where tau is shown to be monomeric) and NMR that a small portion of tau adopts a beta-strand like configuration when monomeric. I am neither an expert in AUC nor NMR, so I need to pose the question of whether this region might be involved in transient dimerization and the beta-strand conformation would thus result from beta-sheets being formed transiently. If the authors can exclude this, then it needs to be discussed. Conversely, if they cannot exclude it, then this possibility needs to be entertained. I do not see such a possibility undercutting any of the conclusions in the paper.

p. 5) A left-handed twist is shown and described, but there is absolutely no explanation as to how the hand was determined.

p. 6) "many of which lack the helical twist required to solve their structures". This appears to be a conceptual error. Imagine a 2-sub-1 screw, which will not display any "twist" along a single filament. This filament is certainly helical, and if all azimuthal orientations of the filament are present in the ice, then all information is present that is needed for a helical reconstruction. But based upon personal experience, both Relion and cryoSPARC will fail using their helical approaches. If one treats this as an asymmetric single-particle and imposes helical symmetry subsequently, such structures can be solved.

p. 9) "Polymorphism is a common phenomenon in crystallography." It is also a common phenomenon in peptide assemblies, some of which have been used as models for amyloid while others have been used for biomaterials (e.g., Wang et al., "Deterministic chaos in the self-assembly of β sheet nanotubes from an amphipathic oligopeptide", Matter, 2021).

Referee #2 (Remarks to the Author):

Disease-specific tau filaments assemble via
polymorphic intermediates

Sofia Lovestam¹, David Li¹, Jane L. Wagstaff¹, Abhay Kotecha², Dari Kimanius¹, Stephen H. McLaughlin¹, Alexey G. Murzin¹, Stefan M.V. Freund¹, Michel Goedert^{1*} & Sjors H.W. Scheres^{1*}

Lovestam et al. apply cryo-EM to image a time series of recombinant tau constructs undergoing early steps of filament formation in vitro. Preliminary studies by solution NMR and Thioflavin T fluorescence enabled identification of a tau segment comprising residues 302-316 which forms the pelletable material after 120 min of centrifugation. Filaments were found. At later time points, numerous types of amyloid filaments were found. Some 163 cryo-EM structures were determined by helical reconstruction in RELION, and 44 different atomic models were built. Tau polymorphs characteristic of filaments extracted from autopsied Alzheimer's and CTE brains were found, respectively in solutions containing 100 mM MgCl₂ (favoring formation of AD PHF polymorphs) and in solutions in which 150 mM NaCl replaces MgCl₂ (favoring CTE polymorphs). At later times (180 minutes) multiple polymorphs of filaments were observed. In the AD conditions some are reminiscent of the C-shaped PHF polymorph, most with two protofilaments. By 720 minutes most filaments adopted the PHF ordered core. In the NaCl (CTE solution) a greater number of intermediate structures formed than in the MgCl₂ solution, but by 720 minutes, filaments were of the CTE types.

The authors describe the numerous filament shapes, and the progression of structural types in terms of models for primary and secondary nucleation and prion-like spreading.

This manuscript describes enormous amounts of data and thoughtful analysis. Particularly notable are:

- The NMR studies of purified tau monomer in solution, pointing to segments with tendency to form extended beta-strand conformations, and correlation times of segments with increased tendency to adopt extended structure;
- Time dependent Thioflavin T studies (by comparison of pre-ThT-treated and post-ThT-treated solutions) confirming that ThT does not affect filament formation, and also that some filament structures do not fluoresce in the presence of ThT;
- The solution conditions that reliably convert a pure construct of monomeric tau into either AD-associated polymorph or the CTE-associated polymorph.
- The profusion of 163 structures of tau filaments with details of determination and of conformation as summarized in Figure 4.
- The time-dependent evolution of filament formation is analyzed in Figure 5, offering molecular models for filament nucleation.

Overall, this is a highly original paper, in which understanding of the time dependence of amyloid structure is elevated to a new level of detail, and fresh ideas about the molecular basis of amyloid nucleation are proposed.

Having said this, there are statements and aspects of figures in the manuscript that are confusing and in some cases misleading, and require adjustment before publication.

1. Precise information for the solution conditions that yield the PHF polymorph of recombinant tau is important. In Lovestam et al. eLife 2022, the concentration of MgCl₂ is given as 200 mM; here as 100 mM. Are both values equally effective? Also, the 2022 paper reports 96% of picked filaments form PHFs at 48 h, where in Figure 4 (upper right-hand corner) > 20% of filaments are PHF at 12 h. Would the conversion to PHFs be near complete at 48h?

2. Page 2, line 5: It seems misleading to assert that "Tau is the only known exception" to the rule that in vitro filaments differ from brain-extracted. In vitro filaments of alpha-synuclein and amyloid-beta also exhibit similarity to brain-extracted filaments. Alpha-synuclein filaments extracted from a patient with juvenile onset synucleinopathy (8bqv) superimpose on recombinant a-syn (6peo) with an RMSD of only 0.9 Å for 357 atom pairs superimposed. Similarly, filaments from a multiple system atrophy patient (6xyo) superimpose on recombinant a-syn (6peo) with an RMS of 1.3 Å for 297 atom pairs. A segment of the filaments from Parkinsons Disease Dementia (8a9l residues 51-66) superimpose on recombinant a-syn (6ufr) with 1.2 Å RMSD for 116 atom pairs. A segment of the filaments from Alzheimers Disease Dementia (7q4m residues 24-42) superimpose on synthetic A-beta (7f29) with RMSD 0.8 Å over 121 atom pairs. To ignore these similarities oversimplifies the state of knowledge about the degree that recombinant proteins reflect ex-vivo proteins. The authors could edit this sentence to reflect the observation that coincidence between brain-extracted and in vitro-produced filament structures is uncommon among currently available data, but the coincidence is not limited to tau.

3. Page 5 last paragraph. This important paragraph is the first description of the structure of the FIA, yet the description of the interactions seems incomplete and partially misleading in terms of the precursor structures. The authors note the hydrophobicity of the interaction interface but not the zippering of sidechains, the hallmark of amyloid. They point out the similarity in structure between FIA and a region of pectate lyase. While perhaps noteworthy, what is the biological significance of this similarity? Are the authors suggesting that this local valine/isoleucine interface is an evolved interaction for a stable interface? They give no evidence of this. Then the authors dismiss the highly similar crystal structure of the exact same VQIVYK sequence of the FIA, as having a major difference from the FIA because the former is less twisted. Yet the similarities are overwhelmingly strong: both are class I steric zippers, with the Val-Ile-Tyr interactions detailed by the authors. Is not the interface of both the FIA and the precursor zipper crystal structure of the same sequence essentially the interaction depicted by the light purple pair and the dark purple pair of segments of the upper left cartoon of Figure 6? Are not the H-bonded relationships of the dark and light purple pairs exactly the same as the crystal VQIVYK crystal structure, except for the twist? When we superimpose 306-VQI-303 from two interfacing chains of the VQIVYK crystal structure as a rigid unit onto the corresponding two chains of FIA, we observe an RMSD of only 0.97 Å (48 atom pairs superimposed). The dry interface structure is conserved despite a 6-degree difference in twist and the presence or

absence of 434 flanking residues. The conservation of structure highlights the robustness of this dry interface, and points to the importance of short sequences in amyloid fibrillogenesis. The truly remarkable message of this story lies in the structural similarities that persist despite different environments, not their differences. In fact, the structural difference incurred by the 6-degree difference in twist has little impact on the free energy of solvation (-7.8 vs -8.1 kcal/mol/chain for FIA versus a manually untwisted FIA structure), inferring that twist plays a relatively insignificant role in the potential of VQIVYK to form amyloid.

4. Figure 4. We applaud the authors for organizing this enormous amount of data into a single comprehensive figure. Clearly, it took much time and effort. There are 57 structures represented, organized by time of appearance (horizontal axis) and abundance (vertical axis). This organization is helpful; however, some aspects are confusing. For example, the same fold seems to be represented at multiple time points in the graph. The small dimensions of the ribbon representations and lack of labeling make it difficult to discern which structures are unique. We suggest that each unique filament structure be represented on a separate horizontal line. The vertical position of the horizontal line could be organized by filament abundance. As in a Gantt chart, a horizontal bar behind the structure could be adjusted in length to indicate the duration of appearance of that structure in the experiments. It would be helpful to include PDB ID code labels for those structures with atomic models and EMD codes for maps without models. Including this additional level of organization would improve the accessibility and impact of this figure.

5. Figure 5 presents the important insight that the folds seem to represent points in a continuous spectrum ranging from 'J' to 'C' shaped. The authors highlight changes in the PGGG regions, but these changes are likely to be reactions to energetically stronger driving forces in the core packing. Can the authors say anything about what drives the changes in the core structure? For example, do the later folds contain fewer voids than the earlier folds? Do the later folds have less exposed hydrophobic surface?

6. Figure 6 is a praise-worthy attempt to depict possible mechanisms of tau filament nucleation, both primary and secondary. The problem is clarity, which can be achieved by expansion of the legend. (Should this all be the subject of a separate article?). The following is our (limited) understanding of line a.

In the primary nucleation step, the light purple VQIVYK segment of the light grey chain forms a homotypic class I steric zipper with an identical light purple VQIVYK segment and the pair recruit an identical layer above. The upper layer is shown in an exploded perspective view with apparently separated dark purple segments—although the two dark purple segments are actually interfacing in the same way as the two light purple segments in the layer below.

In the next step (after the first grey arrow), the VQIVYK segment of another tau molecule hydrogen-bonds to the exposed VQIVYK segment at the tip of the filament, and the C-terminus of this new chain bonds its complementary sequence, in a hetero-zipper to its own VQIVYK. This initiates a tau-amyloid fold. This model of secondary nucleation is new and has a structural basis. (But what is the difference in the species shown before and after the second grey arrow?)

If our understanding is correct, the legend needs more information. If our understanding is incorrect, the legend needs even more information.

Additional labels for the a, b, and c rows of Figure 6 would help. For example, in panel a 'Secondary

nucleation by folding'. In panel b 'Secondary nucleation by sliding', In panel c 'Secondary nucleation by the addition of protofilaments'. In panel d 'Secondary nucleation by conformational change'. The mechanism in panel e seems to be the same as panel c. Both panels show addition of protofilaments to the side of the filament. What is the distinction between mechanisms in panels c and e? The small black arrows of row b are undefined. The legend should specify that gray color indicates disordered structure. This distinction is especially important for comprehending the significance of the structural difference before and after the 3rd arrow in panel a.

Minor comments for consideration of the authors:

1. Abstract. 4 lines from bottom of page: the meaning of 'both reactions' may not be clear to readers; 'reactions' have not been mentioned.
2. Results, line 5: 'increased' compared to what?
3. Page 3 bottom. 'Multiple different protein preparations were used among the replicates.' This sentence seems ambiguous. It could be interpreted to mean that the authors were screening for different protein purification procedures. Perhaps it would be clearer to say that 'Protein samples were prepared at multiple times and the products were considered to be identical.'
4. Figure 2. Can you indicate the radius of the 4.75 Å spacing in each inset?
5. Figure 4 legend. Indicate that the numbers in the pie charts are the replicate numbers.
6. Page 4 middle. The finding that the inflections in Figure 2 panels a and b do not coincide is an important point, but the ability to visualize this idea requires the reader to mentally superimpose the two panels. Consider an additional panel in Figure 2 that shows this superposition of panels a and b. The non-coincidence of the ThT and pelleted mass would be more obvious and emphasize the novelty of examining this previously unexplored, early region of the filamentation pathway.
7. Page 5 middle. The authors claim "The ordered core of the FIA is the smallest of any amyloid observed by cryo-EM reconstruction to date". Please note that the RIPK3 cryo-EM structure has fewer residues in its core: only 22 residues (PDB ID 7da4) compared to 30 in FIA.
8. Page 5 last paragraph. It would be helpful to cite the range of twist angles observed among tau filaments, in order to appreciate how much of an outlier is 6 degrees.
9. Page 6 Paragraph 2. The reference to 24 CryoEM maps as '24 different structures' is confusing. The word 'structure' usually refers to atomic models. Could the authors use 'CryoEM maps' to refer to the maps, to avoid ambiguity of the word 'structure'? The same ambiguity occurs on Page 8 middle.
10. Page 7, last line and Page 8, first line: of course these are 'amino acid residues', or simply 'residues', not 'amino acids'.
11. Page 7, 3 lines from bottom: 'amino- and carboxy-terminal ends'; Max Perutz specifically railed against this usage as redundant. He insisted instead on 'amino- and carboxy ends'.
12. Page 8 middle. Please consider illustrating the density for the ion pair in the CTE fold. How confident are the authors in the assignment of this density to ions, especially when the surrounding environment is presumably hydrophobic?
13. Page 8 bottom. Please give a more quantitative statement than 'ran a bit slower'
14. Page 9 bottom. To render this article accessible to a wider audience, it would be helpful to give your definitions of primary and secondary nucleation. Also, perhaps the authors would comment on the relationship of the models of Figure 5 to the Knowles definition of secondary nucleation.

Signed: David Eisenberg and Michael Sawaya, UCLA

Author Rebuttals to Initial Comments:

We thank the referees for their time and their constructive comments, which we address in blue below.

Referee #1 (Remarks to the Author):

This paper on assembly kinetics of amyloid tau filaments is a tour-de-force and shows how cryo-EM may be used to capture multiple intermediate states on the pathway to mature amyloid filaments. The novel results are very likely to have a broad impact on the understanding of tau neuropathologies and will likely shape a great deal of future work in this area. I had only a number of minor points to raise:

p. 2) It is argued from using AUC (where tau is shown to be monomeric) and NMR that a small portion of tau adopts a beta-strand like configuration when monomeric. I am neither an expert in AUC nor NMR, so I need to pose the question of whether this region might be involved in transient dimerization and the beta-strand conformation would thus result from beta-sheets being formed transiently. If the authors can exclude this, then it needs to be discussed. Conversely, if they cannot exclude it, then this possibility needs to be entertained. I do not see such a possibility undercutting any of the conclusions in the paper.

The NMR data shows that parts of the main chain of tau297-391 monomers adopt extended conformations that are similar to those found in beta-strands. This does not mean they actually form beta-sheets. To make this clearer, we have modified the following sentence:

"Solution-state nuclear magnetic resonance (NMR) confirmed the presence of disordered tau monomers and suggested that amino acids 305-314 and 336-345 have an increased tendency to adopt extended conformations reminiscent of those found in beta-strands."

Both the AUC and the NMR data suggest that, at the concentration used for in vitro assembly, tau297-391 is mostly monomeric. But, the referee is correct in that we can probably not preclude the presence of small amounts of dimers, that might form through inter-molecular beta-sheets. We have therefore also added this sentence:

"Although most tau appears to be monomeric, we cannot exclude the possibility that small amounts of dimers, possibly through transient formation of inter-molecular beta-sheets, are present in solution too."

p. 5) A left-handed twist is shown and described, but there is absolutely no explanation as to how the hand was determined.

This was an oversight. The hand was determined from the chirality of beta-strands when we can see densities for the main-chain oxygens. We have now added the following statement on how the handedness of filaments was determined in the Methods section:

"The handedness of cryo-EM maps with resolutions beyond 2.9 Å was determined from the presence of densities for main-chain carbonyl oxygens. For all other maps, the handedness was determined based on sub-structures that were also present in maps that were solved at resolutions beyond 2.9 Å."

p. 6) "many of which lack the helical twist required to solve their structures". This appears to be a conceptual error. Imagine a 2-sub-1 screw, which will not display any "twist" along a single filament. This filament is certainly helical, and if all azimuthal orientations of the filament are present in the ice, then all information is present that is needed for a helical reconstruction. But based upon personal experience, both Relion and cryoSPARC will fail using their helical approaches. If one treats this as an asymmetric single-particle and imposes helical symmetry subsequently, such structures can be solved.

We have rephrased this to:

"At 140 and 160 min in the PHF reaction, multiple different types of filaments give rise to uninterpretable 2D class averages, many of which lack helical twist (Extended Data Figure 3b). We were unable to solve the structures of these filaments."

p. 9) "Polymorphism is a common phenomenon in crystallography." It is also a common phenomenon in peptide assemblies, some of which have been used as models for amyloid while others have been used for biomaterials (e.g., Wang et al., "Deterministic chaos in the self-assembly of β sheet nanotubes from an amphipathic oligopeptide", Matter, 2021).

We agree with the referee that polymorphism is also common in peptide assembly in general, and amyloid formation in particular. This introductory sentence to the Discussion links our results to Ostwald's rule of stages, which was first described for crystallography.

Referee #2 (Remarks to the Author):

Lovestam et al. apply cryo-EM to image a time series of recombinant tau constructs undergoing early steps of filament formation in vitro. Preliminary studies by solution NMR and Thioflavin T fluorescence enabled identification of a tau segment comprising residues 302-316 which forms the pelletable material after 120 min of centrifugation. Filaments were found. At later time points, numerous types of amyloid filaments were found. Some 163 cryo-EM structures were determined by helical reconstruction in RELION, and 44 different atomic models were built. Tau polymorphs characteristic of filaments extracted from autopsied Alzheimer's and CTE brains were found, respectively in solutions containing 100 mM MgCl₂ (favoring formation of AD PHF polymorphs) and in solutions in which 150 mM NaCl replaces MgCl₂ (favoring CTE polymorphs). At later times (180 minutes) multiple polymorphs of filaments were observed. In the AD conditions some are reminiscent of the C-shaped PHF polymorph, most with two protofilaments. By 720 minutes most filaments adopted the PHF ordered core. In the NaCl (CTE solution) a greater number of intermediate structures formed than in the MgCl₂ solution), but by 720 minutes, filaments were of the CTE types.

The authors describe the numerous filament shapes, and the progression of structural types in terms of models for primary and secondary nucleation and prion-like spreading.

This manuscript describes enormous amounts of data and thoughtful analysis. Particularly notable are:

- The NMR studies of purified tau monomer in solution, pointing to segments with tendency to form extended beta-strand conformations, and correlation times of segments with increased tendency to adopt extended structure;
- Time dependent Thioflavin T studies (by comparison of pre-ThT-treated and post-ThT-treated solutions) confirming that ThT does not affect filament formation, and also that some filament structures do not fluoresce in the presence of ThT;
- The solution conditions that reliably convert a pure construct of monomeric tau into either AD-associated polymorph or the CTE-associated polymorph.
- The profusion of 163 structures of tau filaments with details of determination and of conformation as summarized in Figure 4.
- The time-dependent evolution of filament formation is analyzed in Figure 5, offering molecular models for filament nucleation.

Overall, this is a highly original paper, In which understanding of the time dependence of amyloid structure is elevated to a new level of detail, and fresh ideas about the molecular basis of amyloid nucleation are proposed.

Having said this, there are statements and aspects of figures in the manuscript that are confusing and in some cases misleading, and require adjustment before publication.

1. Precise information for the solution conditions that yield the PHF polymorph of recombinant tau is important. In Lovestam et al. eLife 2022, the concentration of MgCl₂ is given as 200 mM; here as 100 mM. Are both values equally effective? Also, the 2022 paper reports 96% of picked filaments form PHFs at 48 h, where in Figure 4 (upper right-hand corner) > 20% of filaments are PHF at 12 h. Would the conversion to PHFs be near complete at 48h?

The concentration of MgCl₂ was lowered to 100 mM and is as effective as the 200 mM described earlier. Filament percentages are determined slightly differently now - previously percentages were determined as fractions of particles contributing to discernable 2D classes, whereas in this paper they are based on the total number of extracted particles (i.e. also including "bad" particles). Still, waiting longer does increase the yield of PHFs, but it also comes at the cost of filaments clumping together, which complicates cryo-EM analysis. We have included the following sentences of the Methods section to make this clearer:

"AD reactions were performed in 10 mM phosphate buffer at pH 7.2, 100 mM MgCl₂ and 10 mM DTT. Previously, we reported the in vitro assembly of tau(297-391) into PHFs with 200 mM MgCl₂ [20]. The different concentrations of MgCl₂ did not affect the formation of PHFs. CTE reactions were performed in 50 mM phosphate buffer at pH 7.2, 150 mM NaCl and 10 mM DTT. Reactions were performed for 720 min using 200 rpm orbital shaking at 37 °C. Our previous assembly reactions [20] were performed over 48 hours. Waiting longer increases the total amount of PHFs, but also leads to a clumping together of filaments, which complicates cryo-EM analysis."

2. Page 2, line 5: It seems misleading to assert that "Tau is the only known exception" to the rule that in vitro filaments differ from brain-extracted. In vitro filaments of alpha-synuclein and amyloid-beta also exhibit similarity to brain-extracted filaments. Alpha-synuclein filaments extracted from a patient with juvenile onset synucleinopathy (8bqv) superimpose on recombinant a-syn (6peo) with an RMSD of only 0.9 Å for 357 atom pairs superimposed. Similarly, filaments from a multiple system atrophy patient (6xyo) superimpose on recombinant a-syn (6peo) with an RMS of 1.3 Å for 297 atom pairs. A segment of the filaments from Parkinsons Disease Dementia (8a9l residues 51-66) superimpose on recombinant a-syn (6ufr) with 1.2 Å RMSD for 116 atom pairs. A segment of the filaments from Alzheimers Disease Dementia (7q4m residues 24-42) superimpose on synthetic A-beta (7f29) with RMSD 0.8 Å over 121 atom pairs. To ignore these similarities oversimplifies the state of knowledge about the degree that recombinant proteins reflect ex-vivo proteins. The authors could edit this sentence to reflect the observation that coincidence between brain-extracted and in vitro-produced filament structures is uncommon among currently available data, but the coincidence is not limited to tau.

It is true that brain-extracted filaments of alpha-synuclein and amyloid-beta share common substructures with some filaments of recombinant proteins assembled in vitro. These observations have been described in our previous works (Schweighauser et al, Nature 2020; Yang et al, Science 2022; Yang et al, Nature 2022). However, none of these substructures make up 100% of the brain-extracted filament structures. To avoid any danger of misinterpretation, we now write:

“Most in vitro reactions yield filaments with ordered cores that are different in structure from human brain filaments, although in some cases identical substructures have been described [8,10,19]. Only for tau have in vitro assembly conditions been reported that yield filaments that are identical to those derived from human brains.”

3. Page 5 last paragraph. This important paragraph is the first description of the structure of the FIA, yet the description of the interactions seems incomplete and partially misleading in terms of the precursor structures. The authors note the hydrophobicity of the interaction interface but not the zippering of sidechains, the hallmark of amyloid. They point out the similarity in structure between FIA and a region of pectate lyase. While perhaps noteworthy, what is the biological significance of this similarity? Are the authors suggesting that this local valine/isoleucine interface is an evolved interaction for a stable interface? They give no evidence of this.

The side chains of the FIA do not interdigitate, or “zipper”, like the side chains in the steric zippers described in Sawaya et al, Nature 2007. Instead, the side chains of the FIA pack against each other “head-on”, forming a tight hydrophobic packing interface. This type of packing has been termed a beta-sandwich, and was first described in 1981 by Chotia & Janin [PNAS 78:4146-4150] and by Cohen et al [J.Mol.Biol, 183:253-272]. The significance of the pectate lyase structure is that it was the first example of a beta-sandwich made of parallel beta-sheets. The similarity with the FIA does not lie in its primary sequence, and does therefore not suggest an evolutionary origin of the interactions. Instead, the similarity lies in the packing of opposing parallel beta-strands, as highlighted in Extended Data Figure 6a and 6d. The main point is that the FIA packing is not unique, but is like those of globular beta-sandwich proteins.

Then the authors dismiss the highly similar crystal structure of the exact same VQIVYK sequence of the FIA, as having a major difference from the FIA because the former is less twisted. Yet the similarities are overwhelmingly strong: both are class I steric zippers, with the Val-Ile-Tyr interactions detailed by the authors. Is not the interface of both the FIA and the precursor zipper crystal structure of the same sequence essentially the interaction depicted by the light purple pair and the dark purple pair of segments of the upper left cartoon of Figure 6? Are not the H-bonded relationships of the dark and light purple pairs exactly the same as the crystal VQIVYK crystal structure, except for the twist? When we superimpose 306-VQI-303 from two interfacing chains of the VQIVYK crystal structure as a rigid unit onto the corresponding two chains of FIA, we observe an RMSD of only 0.97 Å (48 atom pairs superimposed). The dry interface structure is conserved despite a 6-degree difference in twist and the presence or absence of 434 flanking residues. The conservation of structure highlights the robustness of this dry interface, and points to the importance of short sequences in amyloid fibrillogenesis. The truly remarkable message of this story lies in the structural similarities that persist despite different environments, not their differences. In fact, the structural difference incurred by the 6-degree difference in twist has little impact on the free energy of solvation (-7.8 vs -8.1 kcal/mol/chain for FIA versus a manually untwisted FIA structure), inferring that twist plays a relatively insignificant role in the potential of VQIVYK to form amyloid.

The packings observed in VQIVYK crystal structures represent other examples of close-packed beta-sandwiches. We have now added the following sentence to highlight the similarity between the FIA and one of these crystal structures (2ON9; other crystal structures of VQIVYK, like 4NP8 and 5K7N, display different packings):

“Nevertheless, in one of these crystal structures (PDB-ID 2ON9 [31]), valine 306, isoleucine 308 and tyrosine 310 form a similar hydrophobic close-packed interface as observed in the FIA (Extended Data Figure 6).”

4. Figure 4. We applaud the authors for organizing this enormous amount of data into a single comprehensive figure. Clearly, it took much time and effort. There are 57 structures represented, organized by time of appearance (horizontal axis) and abundance (vertical axis). This organization is helpful; however, some aspects are confusing. For example, the same fold seems to be represented at multiple time points in the graph. The small dimensions of the ribbon representations and lack of labeling make it difficult to discern which structures are unique. We suggest that each unique filament structure be represented on a separate horizontal line. The vertical position of the horizontal line could be organized by filament abundance. As in a Gantt chart, a horizontal bar behind the structure could be adjusted in length to indicate the duration of appearance of that structure in the experiments. It would be helpful to include PDB ID code labels for those structures with atomic models and EMDB codes for maps without models. Including this additional level of organization would improve the accessibility and impact of this figure.

All structures shown in Figure 4 are unique; none of them are represented multiple times. We agree that the small size made it difficult to see the differences in the main chain traces. In the revised version of Figure 4, we have increased the size of the structures by 20%. We have also rearranged the figure to improve overall clarity. We now mention in the legend:

"All structures shown are unique and coloured according to the time point at which they are most abundant, averaged across all replicates."

The suggestion of a Gantt-like chart is a good one. Because of space constraints in Figure 4, we have now inserted a new Extended Data Figure 7 with the average filament abundance of each structure across all replicates for all time points. We have also included the PDB and EMDB codes in this figure.

5. Figure 5 presents the important insight that the folds seem to represent points in a continuous spectrum ranging from 'J' to 'C' shaped. The authors highlight changes in the PGGG regions, but these changes are likely to be reactions to energetically stronger driving forces in the core packing. Can the authors say anything about what drives the changes in the core structure? For example, do the later folds contain fewer voids than the earlier folds? Do the later folds have less exposed hydrophobic surface?

We have now used the Amyloid Illustrator to assess the packing of the different structures in Figure 5. This has resulted in a new Extended Data Figure 8. The packing in the earlier J-shaped protofilaments do contain fewer voids, suggesting that the tighter packing of residues in the C-shaped protofilaments may drive the observed changes. We therefore inserted the following sentence to the main text (and a similar remark for the CTE structures):

"The formation of a tighter packing of residues near the $_{332}PGGG_{335}$ motif in the C-shaped protofilaments compared to the J-shaped protofilaments may drive this conformational change (Extended Data Figure 8a)."

6. Figure 6 is a praise-worthy attempt to depict possible mechanisms of tau filament nucleation, both primary and secondary. The problem is clarity, which can be achieved by expansion of the legend. (Should this all be the subject of a separate article?). The following is our (limited) understanding of line a.

In the primary nucleation step, the light purple VQIVYK segment of the light grey chain forms a homotypic class I steric zipper with an identical light purple VQIVYK segment and the pair recruit an identical layer above. The upper layer is shown in an exploded perspective view with apparently separated dark purple segments—although the two dark purple segments are actually interfacing in the same way as the two light purple segments in the layer below.

The dark-to-light purple shading of the ordered core of the FIA was a bit confusing. We have now made the FIA look more similar to the other structures.

In the next step (after the first grey arrow), the VQIVYK segment of another tau molecule hydrogen-bonds to the exposed VQIVYK segment at the tip of the filament, and the C-terminus of this new chain bonds its complementary sequence, in a hetero-zipper to its own VQIVYK. This initiates a tau-amyloid fold. This model of secondary nucleation is new and has a structural basis. (But what is the difference in the species shown before and after the second grey arrow?)

If our understanding is correct, the legend needs more information. If our understanding is incorrect, the legend needs even more information.

Before the second grey arrow the loop between the two strands of the heteromeric packing is still disordered; after the second grey arrow this loop becomes structured. We now explicitly mention that grey represents disorder in the legend (see below), and we have made the grey part of the structure before the second grey arrow less prominent.

Additional labels for the a, b, and c rows of Figure 6 would help. For example, in panel a 'Secondary nucleation by folding'. In panel b 'Secondary nucleation by sliding', In panel c 'Secondary nucleation by the addition of protofilaments'. In panel d 'Secondary nucleation by conformational change'. The mechanism in panel e seems to be the same as panel c. Both panels show addition of protofilaments to the side of the filament. What is the distinction between mechanisms in panels c and e?

We have added the suggested labels to the legend. Besides examples of "Secondary nucleation by the addition or removal of protofilaments", panel e also shows filaments that split into two smaller filament types. This is now also mentioned in the legend (see below).

The small black arrows of row b are undefined. The legend should specify that gray color indicates disordered structure. This distinction is especially important for comprehending the significance of the structural difference before and after the 3rd arrow in panel a.

The legend now reads:

*"Disordered parts of the molecules are shown in grey; other colours represent ordered structures, with the same colours as in Figure 4. Primary nucleation (pink arrow) of disordered monomers with partially rigid b-strands may lead to formation of the FIA. Subsequent secondary nucleation (grey arrows) may then occur through folding back of the carboxy-terminal domain of tau monomers at the end of, or at defects in, the FIA to form the interface between amino acids 305-316 and 370-380 that remains nearly constant in all intermediate and final filament types. This folding back may then lead to the seed of an early J-shaped protofilament. Possibly, the formation of singlets of J-shaped protofilaments (as observed in the CTE reaction) is followed by packing of a second protofilament to form more stable doublets. **b.** Secondary nucleation by sliding of protofilaments relative to each other (indicated with small black arrows), again at the ends of, or at defects in, filaments may lead to the subsequent formation of more stable structures. **c.** Alternatively, secondary nucleation through the addition and removal of protofilaments may happen when more stable protofilament interactions form at the sides of existing filaments. **d.** Secondary nucleation through conformational change may lead to protofilament maturation, from early J-shaped protofilaments to later C-shaped protofilaments, again at the end of, or at defects in, filaments. **e.** Secondary nucleation through the splitting of filaments may lead to two smaller filament types (e.g. Extended Data Figure 3f). Combined with secondary nucleation through the addition and removal of protofilaments, this may lead to an interplay between multiple filament types."*

Minor comments for consideration of the authors:

1. Abstract. 4 lines from bottom of page: the meaning of 'both reactions' may not be clear to readers; 'reactions' have not been mentioned.

We now refer to these as "assembly reactions".

2. Results, line 5: 'increased' compared to what?

We have removed the word "increased".

3. Page 3 bottom. 'Multiple different protein preparations were used among the replicates.' This sentence seems ambiguous. It could be interpreted to mean that the authors were screening for different protein purification procedures. Perhaps it would be clearer to say that 'Protein samples were prepared at multiple times and the products were considered to be identical.'

Changed as suggested.

4. Figure 2. Can you Indicate the radius of the 4.75 Å spacing in each inset?

Done.

5. Figure 4 legend. Indicate that the numbers in the pie charts are the replicate numbers.

Done.

6. Page 4 middle. The finding that the inflections in Figure 2 panels a and b do not coincide is an important point, but the ability to visualize this idea requires the reader to mentally superimpose the two panels. Consider an additional panel in Figure 2 that shows this superposition of panels a and b. The non-coincidence of the ThT and pelleted mass would be more obvious and emphasize the novelty of examining this previously unexplored, early region of the filamentation pathway.

We have added a faded version of the data in panel a into panel b in both Figure 2 and Extended Data Figure 2 as suggested.

7. Page 5 middle. The authors claim "The ordered core of the FIA is the smallest of any amyloid observed by cryo-EM reconstruction to date". Please note that the RIPK3 cryo-EM structure has fewer residues in its core: only 22 residues (PDB ID 7da4) compared to 30 in FIA.

A good catch! We now write: *"The ordered core of the FIA only comprises amino acids 302GGGSVQIVYKPVDSL316 from two anti-parallel tau molecules, with a predominantly hydrophobic close-packed interface."*

8. Page 5 last paragraph. It would be helpful to cite the range of twist angles observed among tau filaments, in order to appreciate how much of an outlier is 6 degrees.

We have now added the statement: *"(other known tau filaments, including those described in this paper have twists between -1.65° and -0.77°)".*

9. Page 6 Paragraph 2. The reference to 24 CryoEM maps as '24 different structures' is confusing. The word 'structure' usually refers to atomic models. Could the authors use 'CryoEM maps' to refer to the maps, to avoid ambiguity of the word 'structure'? The same ambiguity occurs on Page 8 middle.

We replaced "structures" by "maps" in both places.

10. Page 7, last line and Page 8, first line: of course these are 'amino acid residues', or simply 'residues', not 'amino acids'.

We now use "amino acid residues" in the Abstract once and we have replaced all subsequent instances of "amino acids" with "residues".

11. Page 7, 3 lines from bottom: 'amino- and carboxy-terminal ends'; Max Perutz specifically railed against this usage as redundant. He insisted instead on 'amino- and carboxy ends'.

Done. We would not dare to upset Max. ;-)

12. Page 8 middle. Please consider illustrating the density for the ion pair in the CTE fold. How confident are the authors in the assignment of this density to ions, especially when the surrounding environment is presumably hydrophobic?

Because there are already so many overlapping structures in Figure 5, and the extra density is not changing from one structure to the next, we have not added it to the figure. The evidence comes mostly from separated densities, interpreted as individual K^+ and Cl^- ions in a 1.9 Å reconstruction, and similar non-separated densities for Na^+ and Cl^- pairs in a 2.3 Å map, as described in Lövestam et al, eLife 2022.

13. Page 8 bottom. Please give a more quantitative statement than 'ran a bit slower'

We have replaced this by "*which at 720 min still contained intermediates that were present at 360 min in the other replicates*".

14. Page 9 bottom. To render this article accessible to a wider audience, it would be helpful to give your definitions of primary and secondary nucleation. Also, perhaps the authors would comment on the relationship of the models of Figure 5 to the Knowles definition of secondary nucleation.

We now define "*primary nucleation (i.e. formation of filaments that is independent of the presence of other filaments)*" and "*secondary nucleation (i.e. formation of filaments that depends on the presence of other filaments)*".

Signed: David Eisenberg and Michael Sawaya, UCLA

Reviewer Reports on the First Revision:

Referees' comments:

Referee #1 (Remarks to the Author):

The authors have done a very good job of addressing my minor concerns.

Edward Egelman
University of Virginia

Referee #2 (Remarks to the Author):

This revised version of this groundbreaking paper clarifies nearly all the points we raised. In particular, the revised versions of informative Figures 4 and 6 and their legends are now far clearer.

The single point which we believe remains misleading is the comparison in the second paragraph of page 4 of the FIA structure to previously determined structures. We are glad that the revised text contracted "major difference" to "difference" when referring to twist on page 4. We are also glad that the text added a sentence at the end of the paragraph acknowledging the similarity of the "hydrophobic interface" of FIA and the crystal structure of VQIVYK. But the text still misses the point about the tightness of the interface as being the key attribute of this interface (and a defining feature of the steric zippers of amyloid fibrils).

Instead of first commenting on the highly similar amyloid crystal structure of precisely the same VQIVYK sequence as that which causes the intermolecular binding of the FIA, the text features a non-amyloid globular protein. When the text proceeds to the crystal structure of VQIVYK, it emphasizes the difference in twist, rather than the close similarity of interaction. In fact, "twist" is a soft energetic variable, as the authors elsewhere acknowledge by the range of twists found in amyloid fibrils. That is, the difference in twist is not energetically significant. Rather the distinguishing feature of amyloid is the tightness of its sheet-sheet interface which contributes strongly to stability. In contrast, the exact value of the twist seems insignificant in terms of biology; twist angles are known to vary, even within the same fibril.

It seems misleading to imply that FIA's similarity to pectate lyase is more meaningful than to VQIVYK. The tightness of fit seems the more biologically relevant quality rather than twist. What is of significance to amyloid structures in general is the hydrophobic effect of interaction of apolar surfaces seen in the FIA, the crystal structure of VQIVYK, and in the hetero-zippers in each of the now numerous cryoEM structures of amyloid fibrils.

In the hope that the authors would amend this paragraph in accord with this idea, we illustrate our point of view in the following:

Interdigitation in FIA is evident when viewed perpendicular to the fibril axis (see attached figure, 8ppo-2on9-2pec-zipper01.png). The resemblance to a zipper is obvious here. Arguing more objectively than appearances, one can quantitate the tightness of fit of the sheet-sheet interface with the shape complementarity statistic. These are given in the attached figure for FIA, VQIVYK, and pectate lyase. The FIA peptide (residues 302-312) shows greater shape complementarity (0.80) than the VQIVYK crystal structure (0.72) or the pectate lyase structure (0.67).

The second attached figure shows that the FIA structure has structural/energetic features of steric zippers.

[In the spirit of transparency, it is important for the Editor to know that Reviewers #2 are coauthors of the crystal structure of VQIVYK, the interaction of which we argue is especially relevant to this paper.]

Author Rebuttals to First Revision:

Our responses are in blue below.

This revised version of this groundbreaking paper clarifies nearly all the points we raised. In particular, the revised versions of informative Figures 4 and 6 and their legends are now far clearer.

The single point which we believe remains misleading is the comparison in the second paragraph of page 4 of the FIA structure to previously determined structures. We are glad that the revised text contracted "major difference" to "difference" when referring to twist on page 4. We are also glad that the text added a sentence at the end of the paragraph acknowledging the similarity of the "hydrophobic interface" of FIA and the crystal structure of VQIVYK. But the text still misses the point about the tightness of the interface as being the key attribute of this interface (and a defining feature of the steric zippers of amyloid fibrils).

We now explicitly mention that "valine 306 and isoleucine 308 in the FIA form a similar tightly-packed hydrophobic interface as observed in one of several crystal forms (PDB-ID 2ON9) of the 306VQIVYK₃₁₁ peptide alone."

Instead of first commenting on the highly similar amyloid crystal structure of precisely the same VQIVYK sequence as that which causes the intermolecular binding of the FIA, the text features a non-amyloid globular protein. When the text proceeds to the crystal structure of VQIVYK, it emphasizes the difference in twist, rather than the close similarity of interaction. In fact, "twist" is a soft energetic variable, as the authors elsewhere acknowledge by the range of twists found in amyloid fibrils. That is, the difference in twist is not energetically significant. Rather the distinguishing feature of amyloid is the tightness of its sheet-sheet interface which contributes strongly to stability. In contrast, the exact value of the twist seems insignificant in terms of biology; twist angles are known to vary, even within the same fibril.

We have removed the pectate lyase from the paper and now first comment on the similarity between the peptide crystal and the FIA, before describing the difference in twist and interactions of tyrosine 310. We do think the difference in twist is worth mentioning here, especially as interactions by tyrosine 310 to the backbone of residues 303 and 305 would not be possible in untwisted beta-sheets, and these residues are not present in the VQIVYK crystal.

It seems misleading to imply that FIA's similarity to pectate lyase is more meaningful than to VQIVYK. The tightness of fit seems the more biologically relevant quality rather than twist. What is of significance to amyloid structures in general is the hydrophobic effect of interaction of apolar surfaces seen in the FIA, the crystal structure of VQIVYK, and in the hetero-zippers in each of the now numerous cryoEM structures of amyloid fibrils.

In the hope that the authors would amend this paragraph in accord with this idea, we illustrate our point of view in the following:

Interdigitation in FIA is evident when viewed perpendicular to the fibril axis (see attached figure, 8ppo-2on9-2pec-zipper01.png). The resemblance to a zipper is obvious here. Arguing more objectively than appearances, one can quantitate the tightness of fit of the sheet-sheet interface with the shape complementarity statistic. These are given in the attached figure for FIA, VQIVYK, and pectate lyase. The FIA peptide (residues 302-312) shows greater shape complementarity (0.80) than the VQIVYK crystal structure (0.72) or the pectate lyase structure (0.67).

In summary, we have removed the pectate lyase from Extended Data Figure 6 and we have amended the paragraph in question, which now reads:

“The ordered core of the FIA only comprises residues $_{302}$ GGGSVQIVYKPVDLS $_{316}$ from two anti-parallel tau molecules, with a predominantly hydrophobic close-packed interface. At its centre, the side chains of valine 306 and isoleucine 308 from opposite protofilaments pack against each other and are flanked by the side chain of tyrosine 310. Thereby, valine 306 and isoleucine 308 in the FIA form a similar tightly-packed hydrophobic interface as observed in one of several crystal forms (PDB-ID 2ON9) of the $_{306}$ VQIVYK $_{311}$ peptide alone ^{30,31}. Whereas the β -sheets in the crystal are flat and stabilised by additional crystal contacts, β -sheets in the FIA are twisted and stabilised by additional hydrogen bonds between the hydroxyl group of tyrosine 310 to the backbone groups of glycine 303 and serine 305 (Extended Data Figure 6).”